# PointAD: Comprehending 3D Anomalies from Points and Pixels for Zero-shot 3D Anomaly Detection

**Qihang Zhou[1], Jiangtao Yan[1], Shibo He[1]\*, Wenchao Meng[1], Jiming Chen[1]**
[1] College of Control Science and Engineering, Zhejiang University
[1] {zqhang, jtaoy, s18he, wmengzju, cjm}@zju.edu.cn

## Abstract

Zero-shot (ZS) 3D anomaly detection is a crucial yet unexplored field that addresses scenarios where target 3D training samples are unavailable due to practical concerns like privacy protection. This paper introduces PointAD, a novel approach that transfers the strong generalization capabilities of CLIP for recognizing 3D anomalies on unseen objects. PointAD provides a unified framework to comprehend 3D anomalies from both points and pixels. In this framework, PointAD renders 3D anomalies into multiple 2D renderings and projects them back into 3D space. To capture the generic anomaly semantics into PointAD, we propose hybrid representation learning that optimizes the learnable text prompts from 3D and 2D through auxiliary point clouds. The collaboration optimization between point and pixel representations jointly facilitates our model to grasp underlying 3D anomaly patterns, contributing to detecting and segmenting anomalies of unseen diverse 3D objects. Through the alignment of 3D and 2D space, our model can directly integrate RGB information, further enhancing the understanding of 3D anomalies in a plug-and-play manner. Extensive experiments show the superiority of PointAD in ZS 3D anomaly detection across diverse unseen objects. Code is available at `https://github.com/zqhang/PointAD`

## 1 Introduction

Anomaly detection, a significant field within deep learning, has been widely applied to diverse domains, including industrial inspection [2, 3, 32, 36, 37, 44, 15, 24, 55, 6, 19, 63]. While 2D anomaly detection has been extensively studied by exploring RGB information [23, 56, 50, 51, 8, 29], real-world anomalies typically present themselves with abnormal spatial characteristics. Relying solely on RGB information poses challenges in detecting some anomalies in many cases, e.g., when the defect mimics the appearance of the object's background or foreground, as shown in Figure 1(a). The emerging field of 3D anomaly detection aims to unveil these spatial relations indicative of abnormal patterns [22, 4, 45, 9, 54, 13].

However, current 3D anomaly detection methods typically store normal point features during training and identify anomalies by measuring the distance between the test feature and these stored features [22, 54, 13]. They all depend on the assumption that target point clouds are available and entirely normal. This assumption does not hold in various situations when the training samples in the target dataset are inaccessible due to privacy protection (*e.g.*, involvement of trade secrets) or the absence of target training data (*e.g.*, a new product never seen before) [63]. Figure 1(b) depicts the setting discrepancy between ZS 3D and unsupervised anomaly detection. These methods mentioned above, which detect anomalies by memorizing or reconstructing normal point features, have limitations in generalizing to unseen objects in Figure 1(c). While zero-shot (ZS) anomaly detection has been explored in 2D images [35, 63], ZS 3D anomaly detection remains a research blank. It is a challenging task as ZS

---

*Corresponding authors.

38th Conference on Neural Information Processing Systems (NeurIPS 2024).

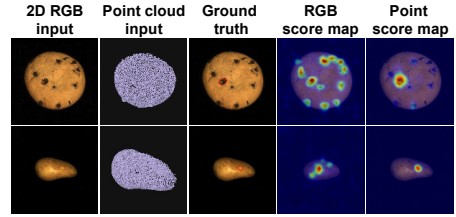
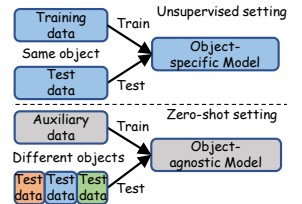
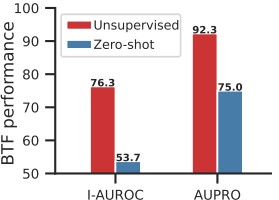

(a) Comparison on anomaly segmentation using different modalities.

(b) Comparison between ZS and unsupervised settings.

(c) BTF Performance degradation on MVTec3D-AD.

Figure 1: Motivation of zero-shot 3D anomaly detection. **(a)**: **Top:** The hole on the cookies presents a similar appearance to the background. **Bottom:** Surface damage on the potato is unapparent to the object foreground. In these cases, leveraging RGB information makes it difficult to detect anomalies that imitate the color patterns of the background or foreground. However, effective recognition can be achieved by modeling the point relations within corresponding point clouds. (b) and (c) depicts the setting difference of ZS and unsupervised manner.

3D anomaly detection necessitates the model to detect 3D anomalies across unseen point clouds with diverse class semantics, requiring a robust generalization capacity in the detection model. Recently, Vision-Language Models (VLMs) with their strong generalization capabilities have been applied to various downstream tasks [40, 61, 41, 49, 25, 26]. Particularly, CLIP has demonstrated its strong ZS performance to detect 2D anomalies [35, 24, 63]. Integrating CLIP into the detection model presents a potential solution to the challenging yet unexplored ZS 3D anomaly detection.

In this paper, we propose a unified framework, namely PointAD, to transfer the knowledge of CLIP to detect 3D anomalies in a ZS manner. PointAD comprehends point clouds from both 3D and 2D: **(1)** deriving 2D representations of point clouds via CLIP by rendering them from multiple views, **(2)** understanding 3D representations by projecting 2D representations back to 3D, and **(3)** enhancing 3D comprehension by additional regularization on 2D representations. After grasping point clouds from points and pixels, we propose hybrid representation learning to capture generic normality and abnormality w.r.t. point and pixel information into learnable text prompts [63]. Specifically, since 3D representation manifests its 2D renderings from different views, we treat each representation as one instance and achieve 3D representation aggregation via multiple instance learning (MIL). On this basis, PointAD explicitly aligns the 2D anomalies, rendered from 3D anomalies, to further enhance 3D understanding. We formulate these 2D anomaly recognition tasks from the multi-task learning (MTL) perspective. PointAD collaboratively learns point and pixel representations, promoting the in-depth understanding of underlying abnormal patterns and thus achieving superior ZS normality and abnormality point recognition. Furthermore, benefiting from collaboration optimization, PointAD can directly integrate additional RGB information and perform ZS multimodal 3D (M3D) detection without extra modules and retraining. The main contributions of this paper are summarized as follows:

- To the best of our knowledge, we are the first to investigate the challenging yet valuable ZS 3D anomaly detection domain. We propose to transfer the strong recognition generalization of CLIP to detect and segment 3D anomalies over diverse objects.

- We introduce a novel ZS 3D anomaly detection approach called PointAD, which provides a unified framework to understand 3D anomalies from points and pixels. Hybrid representation learning is proposed to incorporate the generic normality and abnormality semantics into PointAD, enabling a thorough understanding of 3D anomalies.

- PointAD can incorporate 2D RGB information in a plug-and-play manner for testing. In contrast to other methods that require storing RGB information separately, PointAD offers a unified framework to perform ZS M3D anomaly detection directly.

- Extensive experiments are conducted to demonstrate the superiority of our model in detecting and segmenting 3D anomalies, even outperforming some unsupervised SOTA methods that memorize normal features of target objects in certain metrics. We hope that our model will serve as a springboard for future research on ZS 3D and M3D anomaly detection.

## 2 Related Work

**3D Anomaly Detection** MVTec3D-AD [4], Eyecandies [5], and Real3D-AD [31] provide the point cloud anomalies and the corresponding 2D-RGB information. MVTec3D-AD bridges the connection between 3D and 2D anomaly detection. 3D-ST [4] uses a teacher net to extract dense local geometric

descriptors and design a student net to match such descriptors. AST [45] introduces asymmetric teacher and student net to further improve 3D anomaly detection. IMRNet [28] and 3DSR [58] detect 3D anomalies by reconstruction errors. Instead of only using point clouds, BTF [22], M3DM [54, 53], CPFM [7], and SDM [13] integrate point features and RGB pixel features to detect 3D anomalies. While these approaches exhibit commendable performance by storing object-specific normal point and pixel features within the unsupervised learning framework, such paradigms simultaneously limit their generalization capacity to point clouds from unseen objects, which is crucial to detecting anomalies when the target object is unavailable. To the best of our knowledge, no solution addresses this valuable yet challenging problem. To fill this gap, we introduce PointAD, designed to identify unseen anomalies across diverse objects. PointAD extends CLIP to the realm of ZS 3D anomaly detection and shows robust generalization in capturing generic normality and abnormality within point clouds. Furthermore, PointAD serves as a unified framework, allowing seamless integration of point cloud and RGB modality without additional training.

**3D Feature Extraction**    Conventional methods of 3D feature extraction typically employ a point-based network like PointNet [38] or PointNet++[39] to extract 3D features from point clouds. Alternative approaches convert 3D data into a 2D format [48, 18], enabling 2D image backbones to process 3D information. PointCLIP [59] directly projects raw points onto image planes for efficiency, but this approach causes the produced depth map to lack geometric details. Instead, rendering-based methods [48, 21] generate 2D renderings by rendering point clouds, allowing for better preservation of local semantics. CPFM [7] stores normal features of these 2D renderings for unsupervised 3D anomaly detection. In this paper, we apply this rendering strategy to the source samples to capture generic anomaly semantics for recognizing abnormalities in unseen objects.

**Prompt Learning**    Instead of fine-tuning the whole network, prompt learning just optimizes the model to adapt the network to downstream tasks. CoOp [61, 60] introduces global context optimization to update learnable text prompts for few-shot recognition. DenseCLIP [41] extends it to the dense classifications. More recently, AnomalyCLIP [63] proposes object-agnostic prompt learning to capture the generic normality and abnormality for images. Our model first introduces hybrid representation learning for ZS 3D anomaly detection, enabling the detection of anomalies and abnormal regions.

## 3    PointAD

### 3.1    A Review of CLIP

CLIP, a representative VLM, aligns visual representations to the corresponding textual representations, where an image is classified by comparing the cosine similarity between its visual representation and textual representations of given class-specific text prompts. Specifically, given an image $x_i$ and target class set $\mathcal{C}$, visual encoders output the global visual representation $f_i \in \mathbb{R}^d$ and local visual representations $f_i^m \in \mathbb{R}^{h \times w \times d}$, where $h$, $w$, and $d$ are the height, width, and dimension, respectively. Textual representations $g_c$ are encoded by textual encoder $\mathcal{T}$ with the commonly used text prompt template A photo of a [c], where $c \in \mathcal{C}$. The probability of $x_i$ belongs to $c$ can be computed as:

$$P(g_c, f_i) = \frac{exp(cos(g_c, f_i)/\tau)}{\sum_{c \in \mathcal{C}} exp(cos(g_c, f_i))/\tau)}, \tag{1}$$

where $cos(\cdot, \cdot)$ and $\tau$ represent the cosine similarity and temperature used in CLIP, respectively. The segmentation $S_{i(c)} \in \mathbb{R}^{h \times w}$ for class $c$ can be computed as $Seg(g_c, f_i^m)$, where each entry $(u,v)$ is calculated as $P(g_c, f_{i,u,v}^m)$.

### 3.2    Overview of PointAD

ZS 3D anomaly detection requires a strong generalization capacity to anomalies on unseen objects with diverse object semantics. In this paper, we propose a unified framework, namely PointAD, to detect and segment 3D anomalies in a ZS manner. In Figure 2, PointAD understands point clouds from both pixel and point perspectives. To make CLIP understand 3D point clouds, we first render point clouds from multiple views and extract the pixel representations of these generated 2D renderings via the visual encoder of CLIP. And then, we derive point representations by projecting these pixel representations back to 3D. Learning generic normality and abnormality is significant in recognizing across-object anomalies. We propose hybrid representative learning, which focuses on glocal point and pixel abnormality, to optimize normality and abnormality text prompts, enabling PointAD with strong generalization to identify 3D anomalies on diverse objects. Benefiting from the hybrid representation learning, PointAD can directly incorporate 2D RGB information during testing to achieve ZS M3D detection.

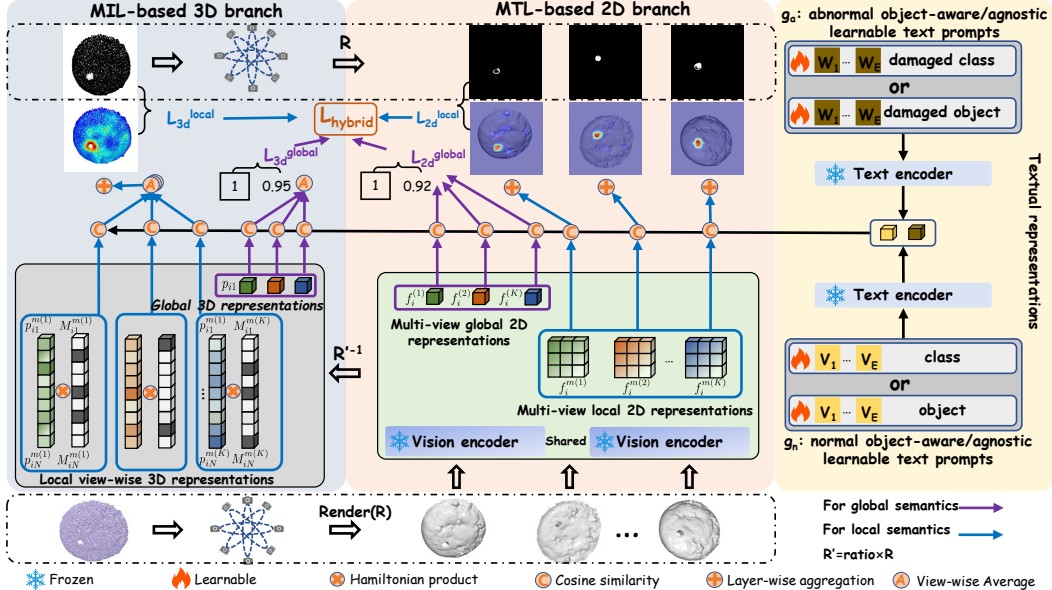

Figure 2: Framework of PointAD. To transfer the strong generalization of CLIP from 2D to 3D, point clouds and corresponding ground truths are respectively rendered into 2D renderings from multi-view. Then, vision encoder of CLIP extracts the renderings to derive 2D global and local representations. These representations are transformed into glocal 3D point representations to learn 3D anomaly semantics within point clouds. Finally, we align the normality and abnormality from both point perspectives (multiple instance learning) and pixel perspectives (multiple task learning) and propose a hybrid loss to jointly optimize the text embeddings from the learnable normality and abnormality text prompts, capturing the underlying generic anomaly patterns.

## 3.3 Multi-View Rendering

Multi-view projection is a crucial technology for understanding point clouds from 2D perspectives. Some multi-view projection approaches project point clouds into various depth maps, providing adequate shape information for class recognition [59]. However, in this paper, our objective is to learn both generic global and local anomaly semantics. Depth-map projection lacks sufficient resolution to represent fine-grained anomaly semantics accurately. Hence, we adopt high-precision rendering to preserve the original 3D information offline. Specifically, given an auxiliary dataset of point clouds $\mathcal{D}_{3d} = \{(x_i^{3d}, y_i^{3d})\}_{i=1}^N$, we define the rendering matrix as $R^{(k)}$ for the $k$-th view, with a total of $K$ views. We simultaneously render point clouds and point-level ground truths from different views to obtain their corresponding 2D renderings, which is given by $x_i^{(k)} = R^{(k)}(x_i^{3d})$ and $y_i^{(k)} = R^{(k)}(y_i^{3d})$, where $x_i^{(k)} \in \mathbb{R}^{H \times W}$ and $y_i^{(k)} \in \mathbb{R}^{H \times W}$ respectively represent the $k$-th 2D renderings and corresponding pixel-level ground truth in the $i$-th point cloud. Note that anomaly pixels are marked as 1, and normal pixels are marked as 0.

## 3.4 Representations for 3D and 2D information

PointAD aims to learn generic anomaly semantics from both 3D and 2D representations, enabling a comprehensive understanding of point and pixel anomaly patterns. For a point cloud $x_i^{3d}$, we first obtain the 2D renderings $\mathcal{X}_i = \{x_i^{(k)}\}_{k=1}^K$. Then, these renderings are encoded via the vision encoder of CLIP to obtain global 2D representations $\mathcal{F}_i = \{f_i^{(k)}\}_{k=1}^K$, and local 2D representations $\mathcal{F}_i^m = \{f_i^{m(k)}\}_{k=1}^K$. As for point cloud representations, we consider that one point cloud will be projected into multiple 2D renderings. Consequently, global 3D representation $p_i$ and local 3D representations $p_i^m$ are expected to include their corresponding 2D representations in each view. Formally, $p_i = \{p_i^{(k)} | p_i^{(k)} = f_i^{(k)}\}_{k=1}^K$ and $p_i^m = \{p_i^{m(k)} | p_i^{m(k)} = \{p_{i,j}^{m(k)}\}_{j=1}^n\}_{k=1}^K$, where $p_{i,j}^{m(k)} = \{p_{i,j}^{m(k)} = f_{i,u,v}^{m(k)}, (u,v) = R'^{(k)}(a_{i,j} | b_{i,j}, c_{i,j})\}$ represents the $j$-th point representation of $i$-th point cloud in the $k$-th view, whose 3D coordinate is $(a_{i,j}, b_{i,j}, c_{i,j})$. $R'^{(k)}$ is the rendering transformation between the point and pixel representation, derived as $R'^{(k)} = \frac{h}{H} R^{(k)}$.

Points at different positions may yield a different number of 2D representations as they are hidden by other points from a specific viewpoint. In this case, we introduce a view-wise visibility mask $M$, where $M_{i,j}^k$ indicates whether the $j$-$th$ point of the $i$-$th$ point cloud is visible in the $k$-$th$ view. We compare the depth of points projected into the same pixel in the same view and set the corresponding visibility mask to 1 for the point with the minimum depth, and to 0 for the other points. Let $\mathcal{Q}_{i,u,v}^{(k)}$ denote the depths set of all points that are projected into the same pixel indexed by $(u, v)$ in the $i$-$th$ point cloud in the $k$-$th$ view. $\mathcal{Q}_{i,u,v}^{(k)}$ and $M_{i,j}^k$ are respectively given as $\mathcal{Q}_{i,u,v}^{(k)} = \{c_{i,j} \mid R'^{(k)}(a_{i,j}, b_{i,j}, c_{i,j}) = (u, v)\}_{j=1}^n$ and $M_{i,j}^{(k)} = \mathbb{I}(i, j, k = \arg\min_{i,j,k}\{c_{i,j} \mid c_{i,j} \in \mathcal{Q}_{i,u,v}^{(k)}\})$, where $\mathbb{I}(\cdot)$ is an indicator function. Local 3D representations of the $i$-$th$ point cloud for the $k$-$th$ view are reformulated as: $p_i^{m(k)} = \{p_{i,j}^{m(k)} * M_{i,j}^{(k)}\}_{j=1}^n$.

## 3.5 Hybrid representation learning

The key of ZS 3D anomaly detection requires the model to capture generic anomaly semantics, rather than relying on specific object semantics. Since CLIP was originally pre-trained to align object semantics, such alignment harms the generalization capacity of CLIP to recognize anomalies on various objects. To adapt CLIP to 3D anomaly detection, we propose a hybrid representation learning, from both 3D and 2D perspectives, to globally and locally optimize textual representations. This enables PointAD to learn more representative text embedding for glocal anomaly semantics alignment. Following previous work [63, 61], we randomly initialize two learnable text templates $t_n$ and $t_a$, in AnomalyCLIP [63] or CoOp manner [61], to obtain more overall text embeddings $g_n$ and $g_a$ to recognize normality and abnormality, respectively.

$$
\begin{aligned}
& t_n = [V_1]\ldots[V_E][object], && t_n = [V_1]\ldots[V_E][class], \\
& \underbrace{t_a = [W_1]\ldots[W_E][damaged][object],}_{\text{PointAD}} && \underbrace{t_a = [W_1]\ldots[W_E][damaged][class],}_{\text{PointAD-CoOp}}
\end{aligned}
$$

where $V$ and $W$ are learnable word embeddings, respectively.

**MIL-based 3D representation learning** To fully incorporate 3D glocal anomaly semantics into PointAD from point information, we respectively devised two losses to capture 3D global anomalies and local anomaly regionals. First, we compute the cosine similarity between the textual representation and its rendering global representations in each view. As point clouds are projected from different views, the resulting renderings in each view reflect certain parts of point clouds. We use view-wise MIL to integrate 2D global representations and then align global labels to capture the global semantics. Formally, the global 3D loss is defined as:

$$
L_{3d}^{global} = \frac{1}{N}\sum_i \text{CrossEntropy}(\frac{1}{K}\sum_{f_i^{(k)} \in p_i} P(g_c, f_i^{(k)}), \max(y_i^{3d})).
$$

As for local point anomaly semantics, we quantify the cosine similarity between textual representations and local representations of 2D renderings. Since points within point clouds are projected from different views, their projections in each view present part characteristics of themselves. We adopt the pixel-wise MIL to achieve the aggregation of point local representation. The point segmentation can be formulated mathematically as follows:

$$
S_{i(a)}^{3d} = \frac{1}{K}\sum_k Seg(g_a, p_i^{m(k)}), S_{i(n)}^{3d} = \frac{1}{K}\sum_k Seg(g_n, p_i^{m(k)}).
$$

However, deriving such 3D segmentation requires similarity computation for each point. It brings a significant memory burden, with a huge computational complexity of $O(Knd)$, which is unaffordable for one NVIDIA RTX 3090 24GB GPU. To address this computational challenge, we resort to the rendering correspondence between points (3D space) and their corresponding pixels within each view (2D space). We first can rewrite 3D segmentation from the view perspective as $S_{i(a)}^{3d} = \frac{1}{K}\sum_k S_{i(a)}^{3d(k)}$. Then, the $k$-$th$ division of 3D segmentation can be transformed into the 2D counterpart through the rendering projection $S_{i(a)}^{3d(k)} = (R^{(k)})^{(-1)} S_{i(a)}^{2d(k)} \otimes M_i^{(k)}$, where $\otimes$ is the Hamiltonian product. The $k$-$th$ 2D counterpart can be computed as $S_{i(a)}^{2d(k)} = \text{Up}(Seg(g_a, f_i^{m(k)}))$, where the operator $\text{Up}(\cdot)$ represents bilinear interpolation from feature space to 2D space. Finally, we can reformulate the 3D segmentation as follows:

$$
S_{i(a)}^{3d} = \frac{1}{K}\sum_k \left((R^{(k)})^{(-1)}\text{Up}(Seg(g_a, f_i^{m(k)})) \otimes M_i^{(k)}\right). \tag{2}
$$

From the equation, we can observe that the primary computation can be conducted in the feature space, with a computational complexity of $O(Khwd)$. This is a substantial overhead reduction compared to $O(Knd)$ since feature space is much smaller than 3D space, *i.e.*, $h \times w \ll n$. In our experiment, $h \times w = 24 \times 24 = 576$, while $n = 336 \times 336 = 112896$. With this transformation, the

Table 1: Performance comparison on ZS 3D anomaly detection in "one-vs-rest" setting.

| Detec. level | Dataset | MVTec3D-AD(10) | | Eyecandies(10) | | Real3D-AD(12) | |
|---|---|---|---|---|---|---|---|
| | Metric | I-AUROC | AP | I-AUROC | AP | I-AUROC | AP |
| | CLIP + R. | 61.2 | 85.8 | 66.7 | 69.2 | 68.8 | 72.3 |
| | Cheraghian | 53.6 | 81.7 | 49.5 | 48.1 | 50.3 | 54.4 |
| | PoinCLIP V2 | 51.2 | 80.1 | 46.1 | 48.1 | 53.1 | 58.1 |
| G. | PointCLIP V2$_a$ | 51.1 | 80.6 | 44.4 | 47.0 | 57.5 | 58.3 |
| | AnomalyCLIP | 56.4 | 83.5 | 57.6 | 59.0 | 55.2 | 57.1 |
| | PointAD-CoOp | 80.9 | 93.9 | 67.7 | 71.8 | 73.9 | 75.9 |
| | PointAD | 82.0 | 94.2 | 69.1 | 73.8 | 74.8 | 76.9 |
| | Metric | P-AUROC | AUPRO | P-AUROC | AUPRO | P-AUROC | AUPRO |
| | CLIP + R. | - | 54.4 | 81.2 | 37.9 | 45.9 | - |
| | Cheraghian | 88.2 | 57.0 | - | - | - | - |
| | PoinCLIP V2 | 87.4 | 52.3 | 43.7 | - | 52.9 | - |
| L. | PointCLIP V2$_a$ | 87.3 | 52.3 | 44.2 | - | 52.2 | - |
| | AnomalyCLIP | 88.9 | 60.9 | 77.7 | 43.4 | 50.3 | - |
| | PointAD-CoOp | 94.8 | 82.0 | 91.5 | 71.3 | 72.6 | - |
| | PointAD | 95.5 | 84.4 | 92.1 | 71.3 | 73.5 | - |

Table 2: Performance comparison on ZS M3D anomaly detection in "one-vs-rest" setting.

| Detec. level | Dataset | MVTec3D-AD(10) | | Eyecandies(10) | |
|---|---|---|---|---|---|
| | Metric | I-AUROC | AP | I-AUROC | AP |
| | CLIP + R. | 60.4 | 86.4 | 73.0 | 73.9 |
| | Cheraghian | - | - | - | - |
| | PoinCLIP V2 | 49.8 | 79.3 | 46.9 | 49.9 |
| MG. | PointCLIP V2$_a$ | 49.4 | 79.8 | 48.5 | 50.5 |
| | AnomalyCLIP | 66.2 | 87.6 | 65.0 | 67.5 |
| | PointAD-CoOp | 83.4 | 94.9 | 73.7 | 76.0 |
| | PointAD | 86.9 | 96.1 | 77.7 | 80.4 |
| | Metric | P-AUROC | AUPRO | P-AUROC | AUPRO |
| | CLIP + R. | - | 56.0 | 78.0 | 31.8 |
| | Cheraghian | - | - | - | - |
| | PoinCLIP V2 | 78.3 | 49.4 | 46.0 | - |
| ML. | PointCLIP V2$_a$ | 79.5 | 51.6 | 46.2 | - |
| | AnomalyCLIP | 91.6 | 70.9 | 85.0 | 56.2 |
| | PointAD-CoOp | 96.5 | 88.8 | 94.9 | 83.6 |
| | PointAD | 97.2 | 90.2 | 95.3 | 84.3 |

entire experiment can be conducted using only a single NVIDIA RTX 3090 24GB GPU. After that, Dice Loss is employed to precisely model the decision boundary of anomaly regions. Let $I$ represent a full-one matrix of the same size as $y_i^{3d}$. Formally, we define 3D local loss $L_{3d}^{local}$ :

$$L_{3d}^{local} = \frac{1}{N} \sum_i \left( \text{Dice}(S_{i(n)}^{3d}, I - y_i^{3d}) + \text{Dice}(S_{i(a)}^{3d}, y_i^{3d}) \right).$$

**MTL-based 2D representation learning**  We further improve PointAD point understanding by capturing 2D glocal anomaly semantics into the object-agnostic text prompt template. We treat the anomaly recognition for one rendering from the point cloud as a task. Hence, we formulate the anomaly semantics learning for multiple 2D renderings as MTL. MTL-based 2D representation learning is divided into two parts for respective alignment to 2D global and local anomaly semantics. For 2D global semantics, we use CrossEntropy to quantify the discrepancy between the textual representations and each global 2D representation. Global MTL-based 2D representation learning $L_{2d}^{global}$ is defined as:

$$L_{2d}^{global} = \frac{1}{NK} \sum_{i,k} \text{CrossEntropy}(P(g_c, f_i^{(k)}), \max (y_i^{(k)})).$$

Also, we focus on 2D abnormal regions to understand pixel-level anomalies. As the anomaly regions are typically smaller than normal regions, we employ Focal Loss to mitigate the class imbalance besides Dice Loss. Let $\oplus$ denote the concatenation operation. Local MTL-based 2D representation learning $L_{2d}^{local}$ is given as follows:

$$L_{2d}^{local} = \frac{1}{NK} \sum_{i,k} \text{Focal}(S_{i(n)}^{2d(k)} \oplus S_{i(a)}^{2d(k)}, y_i^{(k)}) + \text{Dice}(S_{i(n)}^{2d(k)}, I - y_i^{(k)}) + \text{Dice}(S_{i(a)}^{2d(k)}, y_i^{(k)}).$$

### 3.6   Training and Inference

PointAD detects 3D anomalies from both 3D and 2D perspectives and thus combing these above losses to derive hybrid loss $L_{hybrid}$. We minimize $L_{hybrid}$ to incorporate generic anomaly semantics into the text prompt from point and pixel spaces:

$$L_{hybrid} = L_{3d}^{global} + L_{3d}^{local} + L_{2d}^{global} + L_{2d}^{local}.$$

During training, we minimize the hybrid loss $L_{hybrid}$, where the original parameters of CLIP are frozen to maintain its strong generalization. Since our model provides a unified framework to understand anomaly semantics from point and pixel, it can not only perform **ZS 3D anomaly detection** but also **M3D anomaly detection in a plug-and-play way**. Next, we will introduce the inference process in detail:

**ZS 3D/M3D inference**   Given a point cloud $x_i^{3d}$, we regard the 3D segmentation (See Equ. 2) as the anomaly score map: $A_i^m = G_\sigma(S_{i(a)}^{3d})$, where $G_\sigma(\cdot)$ represents the Gaussian filter. The global anomaly score incorporates glocal anomaly semantics and is computed as $A_i^s = \frac{1}{2}(\frac{1}{K} \sum_{f_i^{(k)} \in \mathcal{F}_i} P(g_c, f_i^{(k)}) + \max (A_i^m))$. When the RGB counterpart is available for testing, PointAD could directly integrate RGB information by feeding RGB images to 2D branch to derive 2D representations. We project these 2D representations back to 3D branch to respectively compute RGB anomaly score map and anomaly score as $A_i^{m(rgb)} = P(g_c, f_i^{(rgb)})$ and $A_i^{s(rgb)} = G_\sigma(S_{i(a)}^{3d(rgb)})$. The final multimodal anomaly score map and anomaly score are defined as $A_i^{m(mod)} = \frac{1}{2}G_\sigma(A_i^m + A_i^{m(rgb)})$ and $A_i^{s(mod)} = \frac{1}{2}\left[\frac{1}{2}(A_i^{s(rgb)} + A_i^s) + \max (A_i^{m(mod)})\right]$, respectively.

## 4 Experiment

### 4.1 Experiment Setup

**Dataset**    We evaluate the performance of ZS 3D anomaly detection on three public datasets including, MVTec3D-AD, Eyecandies and Real3D-AD. MVTec3D-AD, Eyecandies, and Real3D-AD are multi-class datasets and respectively contain 10 classes, 10 classes, and 12 classes. Since these training datasets only contain all normal samples, **we use the common zero-shot setting one-vs-rest, where an object test dataset is used to fine-tune PointAD and assess the ZS anomaly detection for the remaining objects.** We also explore a more challenging setting: **cross-dataset ZS generalization, which requires the detection model to generalize to anomalies on other datasets.** For point cloud anomaly detection, we only use point clouds to detect and localize 3D anomalies. In M3D anomaly detection, the 2D RGB information is utilized only for testing. To comprehensively analyze PointAD, we utilize four metrics to assess its performance in both anomaly detection and segmentation.

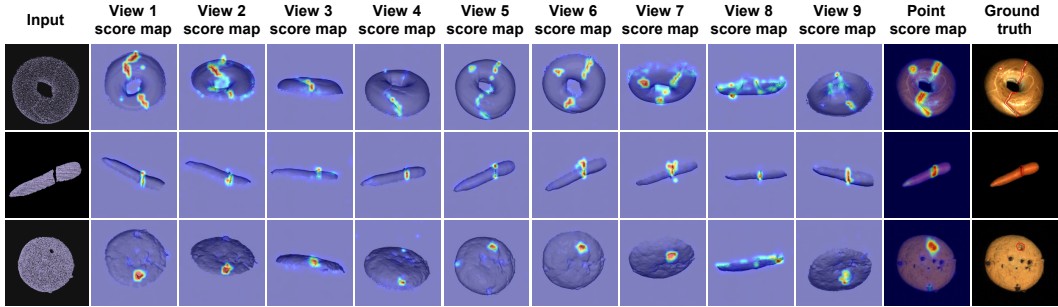

Figure 3: Visualization on anomaly score maps in ZS 3D anomaly detection. Point clouds of diverse objects are input into PointAD to generate 2D and 3D representations. Each row visualizes the anomaly score maps of 2D renderings from different views, and the final point score maps are also presented. More visualizations are provided in Appendix J.

### 4.2 Implementation Details & Baselines

Both point clouds and 2D renderings are resized to $336 \times 336$. We use Open3d library to generate 9 views by rotating point clouds along the X-axis at angles of $\{-\frac{4}{5}\pi, -\frac{3}{5}\pi, -\frac{2}{5}\pi, -\frac{1}{5}\pi, 0, \frac{1}{5}\pi, \frac{2}{5}\pi, \frac{3}{5}\pi, \frac{4}{5}\pi\}$ for most categories. We circularly set the rendering angles, evenly distributing the angles between $-\pi$ to $\pi$. The backbone of PointAD is the pre-trained CLIP model (`VIT-L/14@336px` in `open_clip`). Following [63], we improve the local visual semantics of vision encoder of CLIP without modifying its parameters. During training, we keep all parameters of CLIP frozen and set the learnable word embeddings in object-agnostic text templates to 12. All experiments were conducted on a single NVIDIA RTX 3090 24GB GPU using PyTorch-2.0.0. As there is no work to explore the field of ZS 3D anomaly detection, we make a great effort to provide these comparisons. We apply the original CLIP to our framework for 3D detection, called CLIP + Rendering. Also, we reproduce SOTA 3D recognition works including PointCLIP V2 [64] and Cheraghian [12], and adapt them for ZS 3D anomaly detection. We compare the SOTA 2D anomaly detection approach AnomalyCLIP [63] by fine-tuning it on depth maps. PointAD by default uses object-agnostic text prompts, whereas PointAD-CoOp employs object-aware prompts. Appendix B and C provide more details on implementation and baselines.

### 4.3 Main Results

We fine-tuned PointAD on three objects on MVTec3D-AD, Eyecandies, and Real3D-AD. Over three runs, the averaged results on **one-vs-rest** and **cross-dataset** settings are reported. We use the metric pairs (I-AUROC% % ↑ and AP% % ↑) and (P-AUROC% % ↑ and AUPRO% % ↑) to evaluate the glocal detection performance, respectively. Details of experimental settings see Appendix A. The best and second-best results in ZS are highlighted in Red and Blue. G. and L. represent 3D global and local anomaly detection. M3D global and local anomaly detection are abbreviated as MG. and ML.

**ZS 3D anomaly detection**    Table 1 presents the comparison of ZS 3D performance. Compared to the point-based method Cheraghian and the projection-based method PointCLIP V2, PointAD achieves superior performance on ZS 3D anomaly detection over all three datasets. Especially, it outperforms CLIP + Rendering from 61.2% to 82.0% I-AUROC and from 85.8% to 94.2% AP on MVTec3D-AD. In addition, PointAD achieves superior segmentation performance on ZS 3D anomaly detection, improving MVTec3D-AD by a large margin compared to Cheraghian from 88.2%

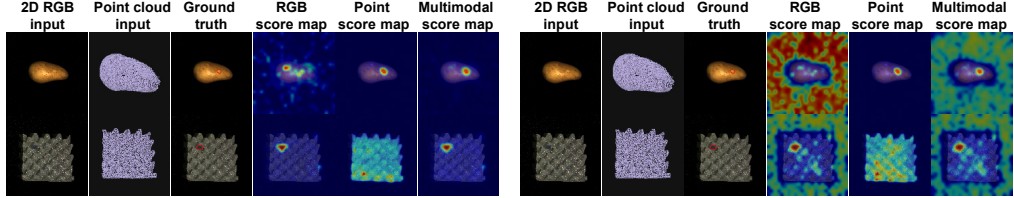

| 2D RGB input | Point cloud input | Ground truth | RGB score map | Point score map | Multimodal score map |

(a) Multimodal visualization with hybrid loss.  (b) Multimodal visualization without 2D glocal loss.

Figure 4: Visualization comparison between PointAD with hybrid loss and without.

to 95.5% P-AUROC and from 57.0 to 84.4% AUPRO. This improvement in overall performance is attributed to PointAD adapting CLIP's strong generalization to glocal anomaly semantics through hybrid representation learning. In addition, PointAD advances PointAD-CoOp across all datasets by blocking the class semantics in text prompts [63].

**ZS M3D anomaly detection** We also compare the ZS M3D anomaly detection when RGB information is available for testing. As shown in Table 2, the results indicate that PointAD can integrate additional RGB information and further boost its performance from 82.0% to 86.9% AUROC and from 94.2% to 96.1% AP for global semantics on MVTec3D-AD. Additionally, as for local semantics, the performance improves from 95.5% to 97.2% P-AUROC and from 84.4% to 90.2% AUPRO. A large performance gain is also obtained on Eyecandies and Real3D-AD. While other methods improve their performance in some metrics, they still suffer from performance degradation in other metrics due to inefficient integration of the two modalities. Instead, PointAD achieves overall improvement across all metrics by incorporating explicit joint constraints on both point and pixel information.

**Cross-dataset ZS anomaly detection** We perform the cross-dataset anomaly recognition to further evaluate the zero-shot capacity of PointAD, where we use one object as the auxiliary and test objects with totally different semantics and scenes in another dataset. We compare all baselines that need fine-tuning. From Table 3 and M3D from

Table 3: Performance comparison on ZS 3D anomaly detection in cross-dataset setting.

| Detec. level | Dataset | Eyecandies(10) | | Real3D-AD(12) | |
|---|---|---|---|---|---|
| | Metric | I-AUROC | AP | I-AUROC | AP |
| G. | PointCLIP V2$_a$ | 45.2 | 48.0 | 57.4 | 58.8 |
| | AnomalyCLIP | 56.3 | 57.1 | 52.7 | 55.7 |
| | PointAD-CoOp | 69.1 | 73.8 | 74.8 | 76.9 |
| | PointAD | 69.5 | 74.3 | 75.9 | 77.9 |
| | Metric | P-AUROC | AUPRO | P-AUROC | AUPRO |
| L. | PointCLIP V2$_a$ | 43.9 | - | 51.9 | - |
| | AnomalyCLIP | 79.6 | 45.4 | 50.3 | - |
| | PointAD-CoOp | 91.8 | 70.5 | 70.1 | - |
| | PointAD | 91.8 | 71.4 | 71.6 | - |

Table 4: Performance comparison on ZS M3D anomaly detection in cross-dataset setting.

| Detec. level | Dataset | Eyecandies(10) | |
|---|---|---|---|
| | Metric | I-AUROC | AP |
| MG. | PointCLIP V2$_a$ | 48.5 | 50.9 |
| | AnomalyCLIP | 65.7 | 68.1 |
| | PointAD-CoOp | 76.3 | 78.9 |
| | PointAD | 78.6 | 80.8 |
| | Metric | P-AUROC | AUPRO |
| ML. | PointCLIP V2$_a$ | 46.3 | - |
| | AnomalyCLIP | 86.2 | 61.3 |
| | PointAD-CoOp | 94.4 | 80.3 |
| | PointAD | 94.0 | 80.7 |

Table 4, PointAD demonstrates strong cross-dataset generalization performance on Eyecandies and Real3D-AD, with nearly no obvious performance decay compared to the one-vs-rest setting. The strong transfer ability highlights its robust generalization capabilities in detecting anomalies in objects with unseen semantics and backgrounds.

### 4.4 Result Analysis

**Visualization analysis.** To intuitively present the strong generalization capacity of our model to unseen anomalies, we visualize the anomaly score maps of the 3D and corresponding 2D counterparts of PointAD on MVTec3D-AD. As shown in Figure 3, PointAD reveals abnormal spatial relationships of points and further captures the generic point anomaly patterns across diverse objects. And, we also visualize the anomaly score map of corresponding 2D counterparts, where 3D point anomalies are transformed into 2D pixel anomalies. It can be observed that PointAD also has a strong detection ability on such 2D anomalies. The strong representative pixel representations from multiple views facilitate more precise 3D anomaly detection. Quantitative results are provided in Section 5. The strong 3D and 2D detection capabilities of PointAD are from hybrid representation learning, which not only enables PointAD to capture the 3D anomalies but also explicitly constrains 2D representations.

**How multimodality makes PointAD accurate.** PointAD is a unified framework that can not only capture point anomalies but also handle 2D information in a plug-and-play manner. As shown in Figure 4(a), we visualize M3D results of PointAD on MVTec3D-AD. The surface damage on the potato presents a similar appearance to the object foreground, which makes it difficult to detect this anomaly with RGB information. On the contrary, the point relations for the color stain on foam are the same as those of normal, but they have a clear distinction in the RGB information. PointAD

can integrate these two modalities, thereby complementing their respective advantages. We further investigate the reason why PointAD can directly leverage both modalities. For this purpose, we experiment without 2D glocal loss. As shown in Figure 4(b), without 2D glocal loss, significant noise disrupts and even covers the RGB score maps, resulting in unpromising multimodal fusions. This illustrates the importance of explicit constraints on the 2D space. Hence, we conclude that the robust multimodal detection capability of our model stems from the collaboration optimization in both 3D and 2D spaces during training. We provide more analysis about failure cases and the computation overhead in Appendix G and H.

## 5 Ablation Study

**Module Ablation.** Here, we investigate the effectiveness of the proposed main technologies by progressively adding the proposed modules. Table 6 illustrates that *vanilla*, which represents the aforementioned CLIP + rendering, performs poor results on both 3D and M3D anomaly detection because CLIP focuses on alignment for 2D object semantics instead of anomaly semantics. With the 3D global branch, we incorporate the global anomaly semantics into PointAD, improving overall performance in local and global detection. After adding the 3D local branch, the performance is further improved, while the pixel-level performance on M3D detection suffers from performance degradation. This is attributed to the absence of 2D constraints, leading to inefficient multimodality fusion as we integrate 2D RGB information in a plug-and-play way. The inclusion of 2D global branch explicitly incorporates 2D anomaly information, which makes PointAD obtain overall performance gain. Finally, by further focusing on 2D anomaly regions, PointAD has a deeper understanding of point clouds from 2D representations and promotes multimodality fusion. Therefore, our model notably boosts the multimodal segmentation performance from 92.9% to 97.2% P-AUROC and from 84.4% to 90.2% AUPRO.

Table 5: Ablation on rendering number.

| View number | Point detection | | Multimodal detection | |
|---|---|---|---|---|
| | Local | Global | Local | Global |
| 1 | (94.1, 79.1) | (72.6, 90.1) | (96.0, 87.6) | (80.4, 93.9) |
| 3 | (95.2, 82.5) | (76.8, 92.1) | (96.9, 89.5) | (83.7, 95.1) |
| 5 | (95.3, 84.3) | (80.8, 93.8) | (97.1, 89.8) | (85.9, 95.7) |
| 7 | (95.3, 84.9) | (81.3, 93.9) | (97.3, 90.0) | (86.5, 95.9) |
| 9 | (95.5, 84.4) | (82.0, 94.2) | (97.2, 90.2) | (86.9, 96.1) |
| 11 | (95.4, 83.8) | (81.7, 94.2) | (97.1, 90.1) | (85.4, 95.5) |

Table 6: Ablation on the proposed modules.

| Module | | | | Point detection | | Multimodal detection | |
|---|---|---|---|---|---|---|---|
| $L_{3d}^{global}$ | $L_{3d}^{local}$ | $L_{2d}^{global}$ | $L_{2d}^{local}$ | Local | Global | Local | Global |
| | | | | (-, 54.4) | (61.2, 85.8) | (-, 56.0) | (60.4, 86.4) |
| ✓ | | | | (91.9, 71.7) | (75.5, 91.9) | (92.6, 81.6) | (80.4, 93.9) |
| ✓ | ✓ | | | (95.2, 82.7) | (81.3, 94.1) | (92.0, 81.4) | (83.9, 95.0) |
| | | ✓ | ✓ | (93.9, 82.8) | (79.3, 91.6) | (91.0, 82.2) | (82.6, 94.1) |
| | ✓ | ✓ | ✓ | (95.5, 84.7) | (81.8, 92.3) | (96.1, 90.6) | (83.7, 95.1) |
| ✓ | ✓ | ✓ | | (95.6, 82.5) | (82.4, 94.5) | (92.9, 84.4) | (85.5, 95.6) |
| ✓ | ✓ | ✓ | ✓ | (95.5, 84.4) | (82.0, 94.2) | (97.2, 90.2) | (86.9, 96.1) |

**View Number Ablation.** PointAD interprets point clouds from 2D renderings, and the quantity of rendering views directly affects the 3D original information acquired by PointAD. Table 5 depicts that the appropriate number of views benefits point understanding from informative views while alleviating negative effects of subpar views. More ablations about the length of learnable prompts, layers of intermediate vision features, and the number of training sets are provided in Appendix F.

## 6 Conclusion

This paper takes the first attempt to study the challenging yet underexplored tasks of ZS 3D and M3D anomaly detection. We propose a unified framework, namely PointAD, to transfer the strong generalization of CLIP to 3D point clouds. PointAD understands 3D anomalies from 3D and 2D spaces. Benefiting from hybrid representation learning, PointAD can recognize generic 3D normality and abnormality across diverse objects and directly integrate RGB information for ZS M3D. Extensive experiments demonstrate the superior ZS detection capacity of our model, whether single modality or multimodality. **Code will be made available once the paper is accepted.**

**Limitations** PointAD utilizes fixed rendering angles to generate 2D renderings across diverse objects. While experimental results demonstrate its superiority, the development of a fine-grained filtering mechanism to select high-quality 2D renderings, particularly for revealing anomalies, remains an avenue for future research.

**Broader Impact** Our paper aims to enhance automated detection and decision-making in smart manufacturing, which does not involve any potential ethical risks. Since the collection of 3D samples is more labor-intensive and costly, our research on using vision-language models for zero-shot point cloud detection can have significant societal impacts, especially in scenarios where target 3D training samples are unavailable due to privacy concerns or the absence of products. We hope that our first exploration of the ZS 3D anomaly detection field could pave the way for further research in this emerging field.

**Acknowledgments**

This work was supported by NSFC U23A20326 and NSFC 62088101 Autonomous Intelligent Unmanned Systems.

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

# A  Dataset

**Dataset**   We evaluate the performance of ZS 3D anomaly detection on three publicly available 3D anomaly detection datasets, MVTec3D-AD, Eyecandies, and Real3D-AD. MVTec3D-AD comprises 4147 point clouds across 10 categories. These objects exhibit diverse object semantics, including bagel, cable gland, carrot, cookie, dowel, foam, peach, potato, rope, and tire. The training dataset comprises 2656 normal point clouds, and the validation dataset comprises 294 normal point clouds. The test dataset includes 948 normal and 249 anomaly point clouds, covering several anomaly types. Point-wise annotations are available for the point clouds. MVTec3D-AD also provides corresponding 2D-RGB image counterparts for the point clouds. We remove the background plane of point clouds in the whole dataset like [22]. Eyecandies also has 10 different classes and provides the corresponding 2D RGB information. Real3D-AD is a recently available dataset, which contains 12 objects. However, it does not provide the RGB information.

**Evaluation Setting and Metric**   Since the training dataset of MVTec3D-AD only contains all normal samples, **we use an object test dataset as the auxiliary dataset to fine-tune PointAD and assess the ZS anomaly detection for the remaining objects.** In particular, we report the average results using different objects as the auxiliary, i.e., carrot, cookie, and dowel for MVTec3D-AD; confetto, LicoriceSandwich, and PeppermintCandy for Eyecandie; seahorse, shell, and starfish for Real3D-AD. Moreover, **we explore more challenging cross-dataset generalization settings, where we use auxiliary data to test all objects of another dataset.** For point cloud anomaly detection, we only use point clouds to detect and localize 3D anomalies in Figure 5(a). In M3D anomaly detection, both point clouds and their 2D-RGB counterparts are utilized, as shown in Figure 5(b). To comprehensively analyze PointAD, we utilize four metrics to assess its anomaly classification and segmentation performance. For anomaly detection, we use the Area Under the Receiver Operating Characteristic Curve (I-AUROC% ↑) and average precision (AP% ↑). Regarding anomaly segmentation, we use point-level AUROC (P-AUROC% ↑) and a restricted metric called AUPRO%(↑) [3] to provide a detailed evaluation of subtle anomaly regions.

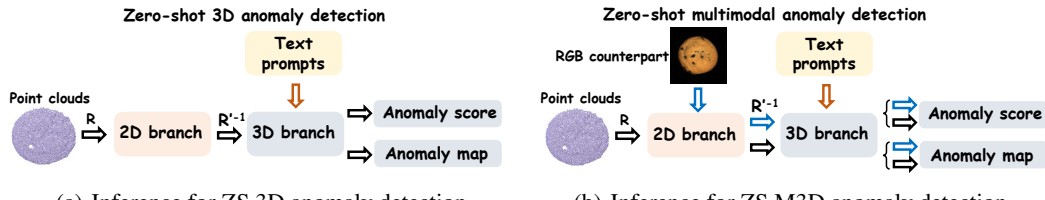

(a) Inference for ZS 3D anomaly detection          (b) Inference for ZS M3D anomaly detection

Figure 5: Inference schematic for ZS 3D and M3D anomaly detection.

# B  Implementation Details

Table 7: Ablation study on the number of rendering views.

| View number | Rendering angles |
|---|---|
| 1 | $0$ |
| 3 | $-\frac{1}{2}\pi, 0, \frac{1}{2}\pi$ |
| 5 | $-\frac{2}{3}\pi, -\frac{1}{3}\pi, 0, \frac{1}{3}\pi, \frac{2}{3}\pi$ |
| 7 | $-\frac{3}{4}\pi, -\frac{1}{2}\pi, -\frac{1}{4}\pi, 0, \frac{1}{4}\pi, \frac{1}{2}\pi, \frac{3}{4}\pi$ |
| 9 | $-\frac{4}{5}\pi, -\frac{3}{5}\pi, -\frac{2}{5}\pi, -\frac{1}{5}\pi, 0, \frac{1}{5}\pi, \frac{2}{5}\pi, \frac{3}{5}\pi, \frac{4}{5}\pi$ |
| 11 | $-\frac{5}{6}\pi, -\frac{2}{3}\pi, -\frac{1}{2}\pi, -\frac{1}{2}\pi, \frac{1}{6}\pi, 0, \frac{1}{6}\pi, \frac{1}{3}\pi, \frac{1}{2}\pi, \frac{2}{3}\pi, \frac{5}{6}\pi$ |

Both point clouds and 2D renderings are resized to $336 \times 336$. We use Open3d library[2] to generate 9 views by rotating point clouds along the X-axis at angles of

[2]https://github.com/isl-org/Open3D

$\{-\frac{4}{5}\pi, -\frac{3}{5}\pi, -\frac{2}{5}\pi, -\frac{1}{5}\pi, 0, \frac{1}{5}\pi, \frac{2}{5}\pi, \frac{3}{5}\pi, \frac{4}{5}\pi\}$ for most categories. Some categories lose their surface completely because they are not stereo point clouds. Table 7 also gives the specific rendering angles of different view number settings. We set the rendering angles in a circular manner, evenly distributing the angles between $-\pi$ to $\pi$. The backbone of PointAD is the pre-trained CLIP model[3] (VIT-L/14@336px). Following [63], we improve the local visual semantics of the vision encoder of CLIP without modifying its parameters and introduce learnable tokens in the text encoder. During training, we keep all CLIP parameters frozen and set the learnable word embeddings in object-agnostic text templates to 12. We use the Adam optimizer with a learning rate of 0.001 to optimize the learnable parameters. The experiment runs for 15 epochs with a batch size of 4. All experiments were conducted on a single NVIDIA RTX 3090 24GB GPU using PyTorch-2.0.0.

## C   Baselines

ZS 3D anomaly detection and M3D anomaly detection have not yet been explored. We first make an adaption for the original CLIP for 3D anomaly detection. Then, we reproduce SOTA ZS 3D classification methods (*i.e.*, PointCLIP V2 [64] and Cheraghian [12]) and adapt them to our settings. SOTA unsupervised 3D anomaly detection approaches are reported as the performance upper bound. All hyperparameters in these baselines are kept the same. We will present the detailed reproduction as follows:

- CLIP + Rendering is a method, where we apply the original CLIP into our framework for ZS 3D anomaly detection. It uses the same rendering procedure as PointAD. Following [24, 63], we integrate anomaly semantics into CLIP by two class text prompt templates: `A photo of a normal [cls]` and `A photo of an anomalous [cls]`, where `cls` denotes the target class name.

- PointCLIP V2 (CVPR 2023) is a SOTA ZS 3D classification method based on CLIP, they project point clouds into depth maps from different views. To adapt PointCLIP V2 into ZS anomaly detection, we replace its original text prompts `point cloud of a big [c]` with normal text prompts `point cloud of a big [c]` and abnormal text prompts `point cloud of a big damaged [c]`.

- AnomalyCLIP (ICLR 2024) is a SOTA zero-shot 2D anomaly detection method. Anomaly-CLIP introduces object-agnostic learning to capture generic anomaly semantics of images. We adapt AnomalyCLIP in 3D detection by fine-tuning AnomalyCLIP on depth images of corresponding point clouds.

- Cheraghian (IJCV 2022) is an approach for ZS 3D classification without foundation models. They directly extract the point presentations by PointNet and use word2vector [34] to generate the textual embedding of an object. To incorporate the anomaly semantics into Cheraghian, we average the textual embeddings of `[c]` and `damaged`. We replace the global representation with dense representations to provide the segmentation to provide the segmentation results.

## D   Related Work

### D.1   2D Anomaly Detection

2D anomaly detection has been studied extensively by leveraging RGB information [47, 46, 14, 52, 42, 11, 62, 43]. Related works can be categorized into two branches: end-to-end and memory-based methods. Representative end-to-end methods exploit knowledge distillation [3, 62, 45] and normalizing flow [20, 57] to model the normal distribution. Instead, memory-based methods [44, 55] store normal features to construct normal prototypes. ZS 2D anomaly detection is proposed to target a challenging problem where training samples are inaccesible [17, 27, 33, 1, 16, 10]. WinCLIP [24] attempts to explore ZS 2D anomaly detection using CLIP. AnomalyCLIP [63] first introduces object-agnostic prompt learning to capture generic normality and abnormality, detecting anomalies across datasets. PromptAD [30] focuses on effectively fusing these embeddings to enhance zero-shot detection performance.

---

[3] https://github.com/mlfoundations/open_clip

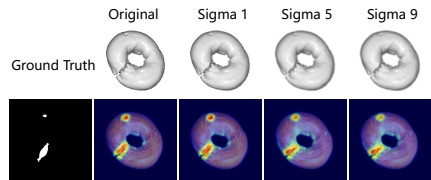

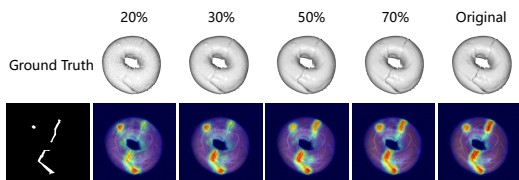

Figure 6: Viusalization with different rendering quality. A larger $\sigma$ represents poorer rendering quality.

Figure 7: Visualization with different resolutions. We downsample entire point clouds with different ratios to obtain diverse resolutions.

Table 8: Analysis on the rendering quality. The original setting is highlighted in gray.

| Blur sigma | Point detection | | Multimodal detection | |
|---|---|---|---|---|
| | Global | Local | Global | Local |
| 0 | (82.0, 94.2) | (95.5, 84.4) | (86.9, 96.1) | (97.2, 90.2) |
| 1 | (80.1, 93.5) | (95.2, 83.2) | (85.5, 95.6) | (97.0, 90.0) |
| 5 | (78.2, 92.5) | (95.1, 82.3) | (83.6, 95.1) | (96.8, 89.7) |
| 9 | (77.6, 92.2) | (95.1, 82.3) | (83.2, 95.0) | (96.5, 89.3) |

Table 9: Analysis on the input solution.

| Downsample ratio | Point detection | | Multimodal detection | |
|---|---|---|---|---|
| | Global | Local | Global | Local |
| 100% | (82.0, 94.2) | (95.5, 84.4) | (86.9, 96.1) | (97.2, 90.2) |
| 70% | (80.6, 93.3) | (95.2, 82.6) | (86.2, 95.1) | (97.0, 90.0) |
| 50% | (77.9, 92.6) | (95.0, 82.0) | (83.6, 94.9) | (96.8, 89.2) |
| 30% | (74.6, 91.1) | (94.9, 81.4) | (81.9, 94.6) | (96.2, 88.7) |
| 20% | (73.9, 90.9) | (94.3, 78.7) | (80.3, 93.4) | (95.3, 87.5) |

# E  Analysis on Rendering Conditions

**Rendering quality**    PointAD interprets point clouds through their corresponding 2D renderings, and the quality of these renderings impacts the information that PointAD can extract from the original point clouds. In our manuscript, we used the Open3D Library to render the point clouds, but it does not provide an API for controlling rendering quality. To simulate varying rendering quality, we applied Gaussian blur with different extents $\sigma$ to the 2D renderings. Sample visualizations are included in Figure 6. Specifically, we conducted experiments on MVTec3D-AD using different blur $\sigma$ values (i.e., ). Table 8 shows that the detection performance of PointAD diminishes as rendering quality decreases (with increasing sigma). However, the degradation is acceptable even when the renderings are heavily blurred ($\sigma$ equals 9). In such cases, PointAD still outperforms baselines that use high-quality renderings.

**Input Resolution**    Here, we study the effect of resolutions of input point clouds. To create low-resolution point clouds, we downsample the entire high-resolution point clouds using Farthest Point Sampling (FPS) with various sampling ratios. This strategy allows us to generate corresponding low-resolution datasets for training and evaluating PointAD. Visualizations of these datasets are provided in Figure 7. We train PointAD using the resulting low-resolution samples and test PointAD on the same resolution. Table 9 demonstrates that PointAD maintains a strong detection capacity for low-resolution point clouds when the downsampling ratios are 20%, 30%, 50%, and 70%. Even at 20% resolution, PointAD still achieves state-of-the-art performance. This indicates that PointAD is generally applicable to point clouds with various resolutions.

**Rendering angles and different numbers of views**    PointAD interprets point clouds from their 2D renderings, where rendering angles and numbers collectively determine the amount of information derived. They have different emphases. For rendering angles, the importance lies in the discrepancy between adjacent angles, as this affects the information granularity that PointAD obtains from adjacent views. When the angle discrepancy is fixed, the number of renderings determines the coverage of 3D information in the resulting 2D renderings. To capture all point cloud information, especially abnormal points, it is crucial to ensure comprehensive coverage. Therefore, our approach in selecting rendering angles and the number of renderings is to guarantee that all points in point clouds are adequately represented.

Based on this principle, we conducted experiments to assess the impact of the number of renderings on PointAD's detection performance, circularly rendering point clouds to ensure even coverage of all points. As shown in Table 5, increasing the number of views allows PointAD to gather more detailed information from the 2D renderings, benefiting from smaller angle discrepancies, which improves detection and localization results. However, when the number of views increased from 9 to 11, we observed a performance decline in PointAD, with the I-AUROC for global multimodal detection dropping from 87.4% to 86.4%. This suggests that incorporating too many views could introduce redundant information, resulting in 2D renderings with extensive overlap and excessive

Table 12: Analysis on the point occlusions.

| Method | Point detection | | Multimodal detection | |
|---|---|---|---|---|
| | Global | Local | Global | Local |
| original | (82.0, 94.2) | (95.5, 84.4) | (86.9, 96.1) | (97.2, 90.2) |
| occlusions | (73.3, 90.6) | (94.3, 80.8) | (83.0, 94.8) | (96.7, 89.5) |

local detail. This overemphasis on local information can impede global recognition. Hence, the appropriate number of views benefits point understanding from informative views while mitigating the adverse effects of redundant local information. To further explore the impact of the absolute angle, we shift the rendering angles while keeping the angle discrepancy unchanged. The original adjacent angle discrepancy in our paper is $\frac{1}{5}\pi$. We divide this discrepancy into four parts and perform angle shifts of $\frac{1}{20}\pi$, $\frac{2}{20}\pi$, and $\frac{3}{20}\pi$ to test the impact of varying rendering angles. Table 10 shows that PointAD maintains consistent performance across different rendering angles, demonstrating its robustness to variations in angles different from those used during training.

Table 10: Analysis on the rendering angle.

| Angle shift | Point detection | | Multimodal detection | |
|---|---|---|---|---|
| | Global | Local | Global | Local |
| 0 | (82.0, 94.2) | (95.5, 84.4) | (86.9, 96.1) | (97.2, 90.2) |
| $\frac{1}{15}\pi$ | (82.6, 94.6) | (95.4, 83.9) | (86.7, 96.0) | (97.1, 90.7) |
| $\frac{2}{15}\pi$ | (82.1, 94.4) | (95.4, 84.1) | (86.4, 95.9) | (97.1, 90.7) |
| $\frac{3}{15}\pi$ | (82.6, 94.6) | (95.4, 84.3) | (86.4, 95.9) | (97.1, 90.7) |

Table 11: Analysis on the rendering lighting.

| Lighting | Point detection | | Multimodal detection | |
|---|---|---|---|---|
| | Global | Local | Global | Local |
| ++ | (82.0, 94.3) | (95.4, 83.7) | (85.7, 95.8) | (97.2, 90.7) |
| + | (82.4, 94.6) | (95.4, 83.8) | (86.1, 95.9) | (97.1, 90.5) |
| original | (82.0, 94.2) | (95.5, 84.4) | (86.9, 96.1) | (97.2, 90.2) |
| - | (82.4, 94.5) | (95.3, 83.9) | (86.4, 95.9) | (97.1, 90.6) |
| -- | (81.9, 94.3) | (95.3, 83.4) | (86.1, 95.8) | (97.1, 90.5) |

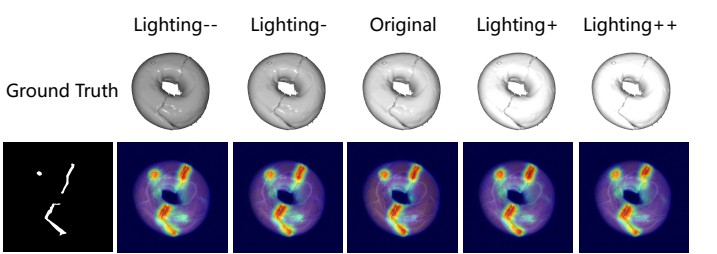

Figure 8: Visualization with different rendering lighting.

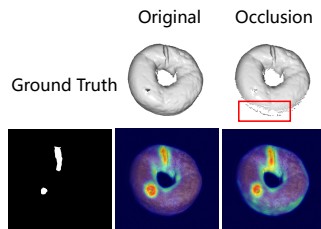

Figure 9: Visualization of occluded point clouds.

**Rendering lighting** Further exploring the robustness of PointAD under different conditions could enhance its generalized detection performance. We conducted ablation studies to test its sensitivity under different lighting conditions and with occluded point clouds below. To evaluate the impact of rendering lighting, we adjusted the lighting conditions to render point clouds, generating variant datasets with different lighting. We used both stronger and weaker lighting to render point clouds compared to the original dataset, covering a broad lighting range. We denote stronger and the strongest lighting as "+" and "++", and weaker and the weakest lighting as "-" and "--". Visualizations of the resulting samples are presented in Figure 8. The experiments were conducted on MVTec3D-AD, where we tested PointAD, trained on the original dataset, on these lighting variant datasets. Table 11 shows that PointAD can still detect anomalies even with significant discrepancies in rendering lighting, suggesting that PointAD is not sensitive to variations in rendering lighting.

**Point occlusions** Next, we evaluated detection performance with occluded point clouds. We occluded point clouds by removing those invisible from a specific rendering angle and then used the same rendering parameters to project the remaining points. During this process, we observed that abnormal regions might be occluded totally. Unlike class classification, where class semantics remain unchanged by point occlusions, removing these anomaly semantics would transform anomaly samples into normal points. Therefore, we selected rendering angles that allow visibility of part or all anomalous regions when the point cloud is an abnormal instance. The occluded point clouds are shown in Figure 9. We then used PointAD, trained on the original dataset, to test the resulting occluded dataset. Table 12 shows that PointAD suffers from performance degradation when the point clouds are occluded. We attribute this to two aspects: 1) Despite this strategy, the occluded

point clouds could still lose part of the anomaly semantics; 2) Occluded point clouds could create unexpected sinkholes on the surface, which may cause PointAD to identify these areas as hole anomalies incorrectly.

# F   Hyperparameter Ablation

Table 13: Ablation study on the length of the learnable prompt.

| Length of learnable prompt | Point detection | | Multimodal detection | |
|---|---|---|---|---|
| | Pixel level | Image level | Pixel level | Image level |
| 6 | (94.6, 83.4) | (81.7, 94.2) | (96.5, 89.8) | (86.6, 96.0) |
| 8 | (95.2, 83.6) | (82.0, 94.2) | (96.8, 90.0) | (86.6, 95.8) |
| 10 | (95.3, 84.0) | (81.8, 94.3) | (97.0, 90.1) | (86.5, 96.0) |
| 12 | (95.5, 84.4) | (82.0, 94.2) | (97.2, 90.2) | (86.9, 96.1) |
| 14 | (95.4, 83.7) | (81.4, 94.1) | (96.9, 89.7) | (85.5, 95.6) |
| 16 | (95.1, 83.0) | (81.5, 94.1) | (96.9, 89.8) | (84.7, 95.6) |

**Length of learnable prompts**   We study the sensitivity of important hyperparameters in PointAD. First, we explore the length of learnable text prompt templates, as shown in Table. 13. As the length of word embeddings increases, PointAD can better learn 3D and 2D anomaly semantics to improve its performance. Nonetheless, with a further increase in length (i.e., from 12 to 16), a decline in performance becomes noticeable. The excessive or insufficient number of learnable word embeddings can lead to performance degradation. An appropriate length (*i.e.*, 12) is important for PointAD to attain comprehensive performance in both 3D and M3D anomaly detection.

**Training set**   We have increased the test data for each category to incorporate more instances on MVTec3D-AD. Originally, we evaluated the rest of the data using only one category as the test data, such as carrot, cookie, and dowel. Now, we attempt to incorporate more instances. Here, we utilized two categories as auxiliary data, including carrot and cookies, carrot and dowel, and cookie and dowel. To further analyze the effect of the size of auxiliary data, we utilized three categories as auxiliary data, selecting all possible combinations from the four sets: carrot, cookie, bagel, and dowel. We present the performance averaged across all groups below. From Table 14 and Table 15, PointAD can incorporate more knowledge about abnormality and normality, improving point detection and multimodal detection performance. Specifically, from one category to three categories, PointAD exhibits improved performance, with P-AUROC increasing from 95.5%, 96.1%, to 96.3%, and AUPRO increasing from 84.4%, 86.3%, to 86.5%. Moreover, I-AUROC increases from 97.2%, 97.5%, to 97.8%, and AP increases from 90.2%, 91.8%, to 92.0%. This trend is also observed in global anomaly semantics.

Table 14: Ablation study on training set size for point detection

| | Training set | Bagel | Cable gland | Carrot | Cookie | Dowel | Foam | Peach | Potato | Rope | Tire | Mean |
|---|---|---|---|---|---|---|---|---|---|---|---|---|
| G. | one | (98.3, 99.6) | (53.7, 86.0) | (97.9, 99.6) | (92.1, 97.9) | (72.2, 92.0) | (69.5, 91.2) | (91.5, 97.6) | (98.8, 99.7) | (91.5, 96.7) | (54.1, 82.1) | (82.0, 94.2) |
| | two | (99.0, 99.8) | (52.5, 85.3) | (98.3, 99.7) | (91.8, 97.7) | (70.2, 91.4) | (68.9, 90.8) | (91.5, 97.6) | (99.2, 99.8) | (90.8, 96.5) | (56.9, 85.0) | (81.9, 94.4) |
| | three | (100, 100) | (52.2, 84.8) | (98.7, 99.7) | (94.2, 98.4) | (71.2, 91.6) | (68.9, 91.1) | (92.6, 98.0) | (99.6, 99.9) | (88.9, 95.8) | (54.5, 83.6) | (82.1, 94.3) |
| L. | one | (98.4, 96.9) | (93.5, 79.5) | (99.4, 96.4) | (87.5, 75.4) | (95.5, 75.2) | (86.5, 54.1) | (99.5, 98.3) | (99.9, 99.1) | (99.3, 89.9) | (95.3, 79.7) | (95.5, 84.4) |
| | two | (99.1, 98.0) | (93.5, 79.9) | (99.6, 97.7) | (90.4, 83.8) | (95.6, 75.2) | (88.3, 57.3) | (99.6, 98.6) | (99.9, 99.4) | (99.5, 91.0) | (96.0, 81.7) | (96.1, 86.3) |
| | three | (99.3, 98.2) | (93.5, 79.8) | (99.6, 97.8) | (92.4, 87.7) | (95.7, 75.7) | (87.9, 57.1) | (99.7, 98.7) | (99.9, 99.4) | (99.4, 90.6) | (95.7, 80.3) | (96.3, 86.5) |

Table 15: Ablation study on training set size for multimodal detection.

| | Training set | Bagel | Cable gland | Carrot | Cookie | Dowel | Foam | Peach | Potato | Rope | Tire | Mean |
|---|---|---|---|---|---|---|---|---|---|---|---|---|
| MG. | one | (98.8, 99.7) | (79.9, 94.7) | (95.5, 98.9) | (86.2, 95.5) | (98.5, 90.6) | (84.4, 96.1) | (96.6, 99.1) | (90.7, 97.0) | (93.6, 97.3) | (74.6, 92.0) | (86.9, 96.1) |
| | two | (98.6, 99.7) | (80.8, 94.8) | (98.3, 99.6) | (85.6, 95.4) | (64.7, 88.9) | (84.9, 96.3) | (95.8, 99.0) | (90.6, 96.8) | (94.5, 97.8) | (77.2, 93.0) | (87.1, 96.1) |
| | three | (98.6, 99.7) | (80.1, 94.5) | (97.3, 99.4) | (91.7, 97.7) | (66.8, 89.5) | (87.3, 97.0) | (97.4, 99.4) | (92.0, 97.4) | (95.0, 98.0) | (74.0, 92.3) | (88.0, 96.5) |
| ML. | one | (99.6, 90.1) | (96.7, 97.9) | (99.4, 85.5) | (92.6, 85.4) | (96.1, 74.0) | (92.4, 98.3) | (99.4, 98.9) | (99.8, 92.9) | (98.8, 87.9) | (97.5, 91.1) | (97.2, 90.2) |
| | two | (99.6, 98.7) | (97.0, 91.6) | (99.4, 97.6) | (93.8, 88.4) | (95.7, 83.5) | (93.9, 78.8) | (99.5, 98.5) | (99.8, 99.3) | (98.6, 90.6) | (98.1, 90.7) | (97.5, 91.8) |
| | three | (99.7, 98.8) | (97.2, 91.1) | (99.5, 98.0) | (95.4, 90.0) | (95.9, 86.1) | (93.8, 77.1) | (99.6, 98.6) | (99.8, 99.2) | (98.6, 91.9) | (98.1, 89.0) | (97.8, 92.0) |

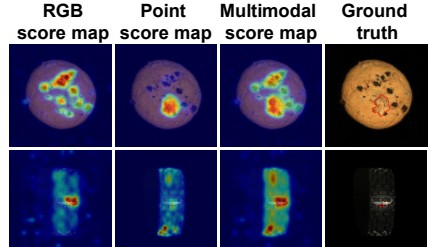

Figure 10: Failure case in PointAD.

Table 16: Effect on point density gap.

| Downsample ratio | Point detection | | Multimodal detection | |
|---|---|---|---|---|
| | Global | Local | Global | Local |
| 100% | (82.0, 94.2) | (95.5, 84.4) | (86.9, 96.1) | (97.2, 90.2) |
| 70% | (81.3, 94.0 | (94.9, 82.6) | (85.6, 95.7) | (96.8, 90.0) |
| 50% | (79.6, 93.4) | (94.7, 81.8) | (84.9, 95.5) | (96.8, 89.9) |
| 30% | (76.6, 91.8) | (94.5, 79.9) | (83.5, 95.1) | (96.4, 89.5) |
| 20% | (72.7, 90.5) | (94.2, 78.2) | (82.0, 94.6) | (95.2, 88.6) |

## G   Failure Cases

In this section, we present the failure cases of PointAD, which we attribute to direct multimodality fusion. Since our model uses hybrid loss to incorporate the 3D and 2D anomaly semantics, it performs ZS multimodality 3D anomaly detection in a plug-and-play manner. However, when one modality prediction deviates severely from the ground truth in rare instances, direct fusion may result in an unpromising multimodal score map. As shown in Figure 10, the hole in the cookie is visually similar to the chocolate on cookies, making it challenging to differentiate the hole anomaly via color information alone. Although PointAD can detect the hole based on its abnormal point relations, the RGB score map heavily influences the final multimodal score map. Conversely, the tire presents an inverse situation where RGB can effectively predict the anomalies, but the point score map fails to recognize it. The false detection could arise from unusual point density and distribution. To demonstrate the effect of point density, we randomly select normal regions of point clouds and subsequently increase or decrease the density of these regions through upsampling and downsampling. We provide a qualitative analysis in 11. The visualization shows that PointAD effectively resists noise at reasonable levels. However, when noise levels are extremely low or high, the corresponding regions become excessively sparse or dense. This causes normal regions to appear similar to hole anomalies or squeezed anomalies, leading PointAD to classify these noisy areas as anomalies. Non-parametric score alignment and filter methods could be a potential direction, which we leave for further work.

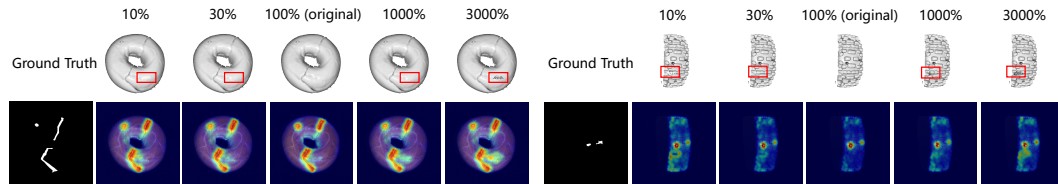

Figure 11: The impact of noise level. We randomly downsample and upsample part of normal regions to create different point densities.

**Robustness to point density**   To investigate the impact of point density differences between the training and test datasets, we train PointAD using high-density point clouds (original dataset) and then test it on the low-density versions of the datasets, downsampled as described in **Input resolution**. Table 16 shows that PointAD can still detect anomalies even when retaining 50% of the points from the original point clouds. However, when more points are removed (30% and 20% sample ratio), PointAD experiences an obvious performance degradation. We attribute the misdetection to the overly sparse point clouds forming holes. Nevertheless, PointAD can still detect anomalies even with

Table 17: Comparison of computation overhead with SOTA approaches on MVTec3D-AD. The unsupervised method is abbreviated as Un.

| Methods | Inference time (s) | FPS | GPU memory usage (Peak) | Point detection Global | Point detection Local | Multimodal detection Global | Multimodal detection Local |
|---|---|---|---|---|---|---|---|
| BTF (Un.) | 0.18 | 5.56 | 1934 | (76.3, 91.8) | (97.6, 92.3) | (89.8, 96.7) | (99.5, 97.1) |
| SDM (Un.) | 0.14 | 7.14 | 2716 | (96.7, 90.9) | (97.0, 91.8) | (92.1, 97.6) | (93.3, 98.1) |
| M3DM (Un.) | 2.86 | 0.35 | 7494 | (85.6, 93.4) | (92.8, 91.6) | (93.2, 92.7) | (98.4, 95.1) |
| CPFM (Un.) | 0.22 | 4.55 | 1379 | (94.9, 98.7) | (97.6, 92.5) | (-,-) | (-,-) |
| 3DSR (Un.) | 0.09 | 11.11 | 3067 | (95.1, 94.3) | (93.8, 91.2) | (97.3, 98.6) | (99.3, 97.4) |
| PointCLIP V2 (ZS) | 1.52 | 0.66 | 9747MB | (78.3, 49.4) | (87.4, 52.3) | (49.8, 79.3) | (51.2, 80.1) |
| CLIP + Rendering (ZS) | 0.27 | 3.61 | 3685MB | (-, 54.4) | (61.2, 85.8) | (-, 56.0) | (60.4, 86.4) |
| Cheraghian (ZS) | 0.35 | 2.86 | 4847MB | (53.6, 81.7) | (88.2, 57.0) | (-, -) | (-, -) |
| WinCLIP (ZS) | 0.29 | 3.45 | 3914MB | (45.2, 77.9 ) | (85.8, 59.4) | (38.7, 74.1) | (87.5, 64.2) |
| AnomalyCLIP (ZS) | 0.19 | 5.26 | 3348MB | (56.4, 83.5) | ( 88.9, 60.9) | ( 66.2, 87.6) | ( 91.6 70.9) |
| Ours (ZS) | 0.40 | 2.52 | 4275MB | (82.0, 94.2) | (95.5, 84.4) | (86.9, 96.1) | (97.2, 90.2) |

a significant gap in point density between the training and test domains (e.g., 100% vs. 20%). This demonstrates PointAD's ability to generalize across different point densities.

# H  Complexity analysis

In Table 17, we provide a comparison of computation overhead among unsupervised and zero-shot manners [4]. The evaluation includes inference time per image, frames per second (FPS), and GPU memory consumption with a batch size of 1. For a fair comparison, we keep NVIDIA RTX 3090 24GB GPU free until we conduct experiments. Compared to PointCLIP V2, our model requires less time to infer an image, achieving higher FPS (2.52 vs. 0.66) with lower graphic memory usage (as discussed in Section 3.5). While CLIP + rendering has a slight advantage in computation overhead, our detection performance significantly outperforms it. Therefore, PointAD achieves a favorable trade-off between performance and computation overhead.

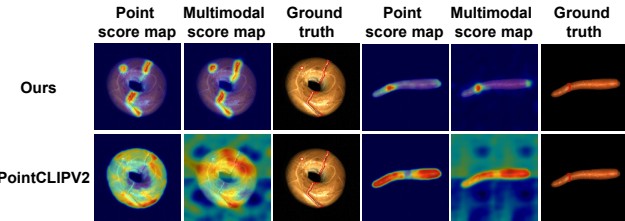

Figure 12: Visualization comparison between PointAD and PointCLIP V2.

# I  Visualization Comparison

To provide intuitive results, we compare the visualization of PointAD with PointCLIP V2. In Figure 12, our model achieves accurate ZS 3D detection through the point cloud. Moreover, given RGB counterparts, PointAD further improves its detection capacity in M3D detection. However, PointCLIP V2 exhibits noisy activations for normal regions. After incorporating RGB information, PointCLIP V2 appears to struggle to fuse these two modalities in a plug-and-play manner, unlike PointAD.

# J  Additional Visualization

We respectively visualize the 2D renderings and corresponding 2D ground truths, which are rendered from 3D pint clouds and ground truths, as shown in Figure 13. We also supplement more zero-shot segmentation results of PointAD in Figure 14.

---

[4]Since the official code for WinCLIP is not available, we reproduce its results using the implementation found at `https://github.com/zqhang/Accurate-WinCLIP-pytorch`

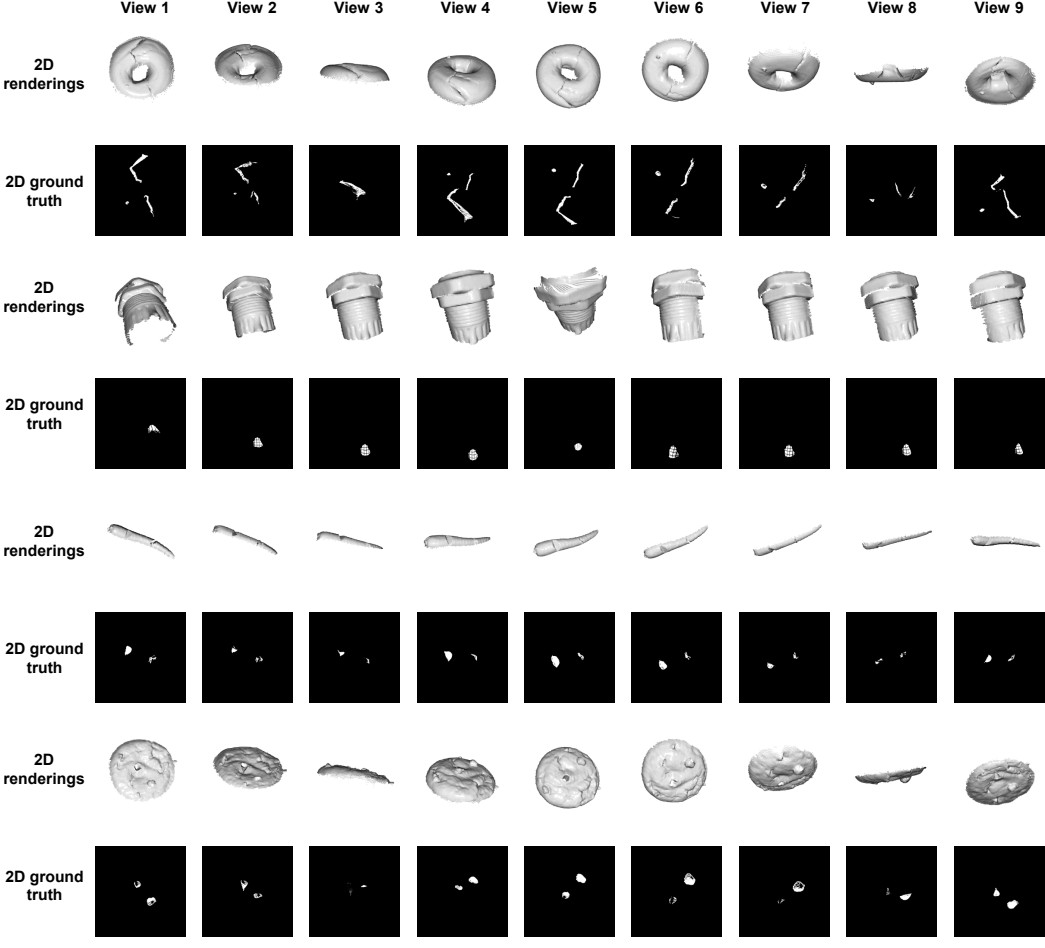

Figure 13: Visualization about 2D renderings and ground truth from different views ($K = 9$).

## K  Detailed results

- We provide the class-level ZS 3D results on MVTec3D-AD in Table 18.
- We provide the class-level ZS M3D results on MVTec3D-AD in Table 19.
- We provide the class-level ZS 3D results on Eyecandies in Table 20.
- We provide the class-level ZS M3D results on Eyecandies in Table 21.
- We provide the class-level ZS 3D results on Real3D-AD in Table 22.
- We provide the class-level ZS cross-dataset 3D results on Eyecandies from MVTec3D-AD in Table 23.
- We provide the class-level ZS cross-dataset 3D results on Eyecandies from MVTec3D-AD in Table 24.
- We provide the class-level ZS cross-dataset 3D results on Real3D-AD from MVTec3D-AD in Table 25.

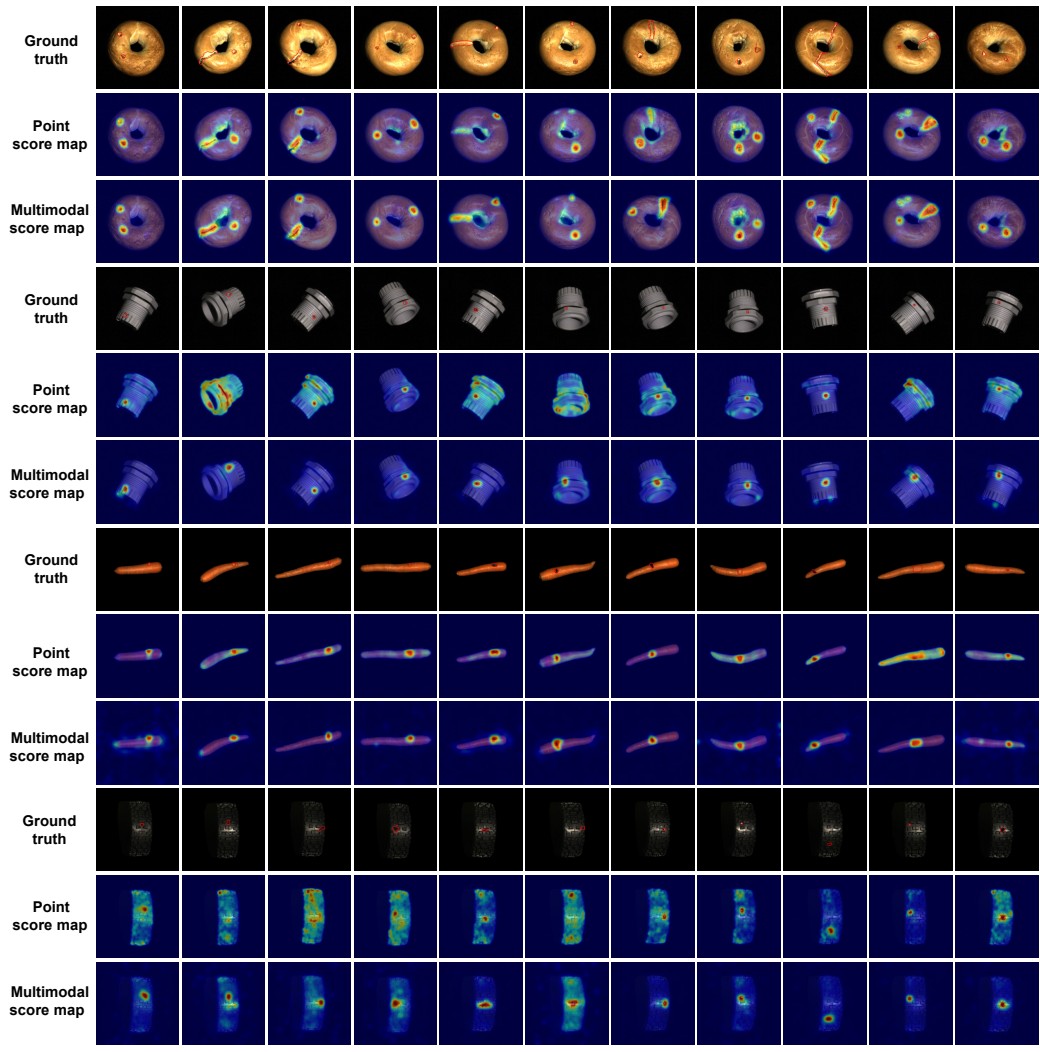

Figure 14: Visualization of point and multimodal score maps in PointAD, which is pre-trained on cookie object.

Table 18: Performance comparison on ZS 3D anomaly detection. The best and second-best results in ZS are highlighted in red and blue. G. and L. represent the global and local anomaly detection.

| | Method | Bagel | Cable gland | Carrot | Cookie | Dowel | Foam | Peach | Potato | Rope | Tire | Mean |
|---|---|---|---|---|---|---|---|---|---|---|---|---|
| G. | CLIP + R. | (53.4, 85.2) | (49.6, 83.2) | (62.9, 89.9) | (65.0, 88.0) | (65.3, 89.3) | (53.0, 78.9) | (72.0, 89.2) | (58.5, 83.4) | (80.0, 90.7) | (52.4, 80.1) | (61.2, 85.8) |
| | Cheraghian | (49.3, 80.5) | (47.1, 80.1) | (52.7, 83.8) | (54.4, 83.0) | (43.3, 78.6) | (47.4, 80.6) | (50.0, 80.5) | (59.7, 84.4) | (72.8, 85.2) | (59.8, 80.5) | (53.6, 81.7) |
| | PoinCLIP V2 | (71.7, 35.9) | (68.6, 39.2) | (94.3, 83.6) | (69.8, 28.5) | (75.5, 47.7) | (67.1, 51.4) | (69.7, 36.5) | (84.6, 57.6) | (91.8, 76.1) | (89.8, 67.5) | (51.2, 80.1) |
| | PointCLIP V2$_\alpha$ | (47.1, 80.7) | (55.1, 84.7) | (47.7, 80.8) | (50.1, 79.8) | (50.9, 82.9) | (57.4, 83.7) | (52.4, 83.5) | (48.2, 78.5) | (54.8, 74.4) | (47.8, 76.8) | (51.1, 80.6) |
| | AnomalyCLIP | (62.8, 86.9) | (51.2, 82.2) | (51.9, 84.4) | (64.9, 86.2) | (50.0, 80.3) | (42.4, 80.1) | (69.4, 90.8) | (61.5, 85.7) | (62.4, 81.5) | (47.8, 77.1) | (56.4, 83.5) |
| | PointAD-CoOp | (98.3, 99.6) | (53.8, 85.7) | (93.2, 98.5) | (89.5, 97.1) | (66.3, 89.8) | (70.3, 91.3) | (89.1, 96.9) | (97.8, 99.5) | (91.1, 96.5) | (59.2, 83.7) | (80.9, 93.9) |
| | PointAD | (98.3, 99.6) | (53.7, 86.0) | (97.9, 99.6) | (92.1, 97.9) | (72.2, 92.0) | (69.5, 91.2) | (91.5, 97.6) | (98.8, 99.7) | (91.5, 96.7) | (54.1, 82.1) | (82.0, 94.2) |
| L. | CLIP + R. | (-, 22.2) | (-, 67.5) | (-, 77.4) | (-, 6.7) | (-, 65.6) | (-, 37.4) | (-, 38.8) | (-, 77.2) | (-, 72.2) | (-, 79.4) | (-, 54.4) |
| | Cheraghian | (73.2, 13.8) | (93.0, 75.6) | (83.9, 45.6) | (82.0, 39.8) | (93.6, 67.0) | (84.4, 45.9) | (84.5, 40.1) | (95.7, 77.1) | (96.2, 74.9) | (95.9, 89.8) | (88.2, 57.0) |
| | PoinCLIP V2 | (78.2, 36.0) | (90.9, 65.7) | (96.4, 76.0) | (74.7, 24.4) | (93.8, 66.0) | (75.1, 18.9) | (86.0, 43.4) | (92.8, 60.1) | (95.6, 71.7) | (90.3, 63.3) | (87.4, 52.3) |
| | PointCLIP V2$_\alpha$ | (79.4, 38.0) | (91.5, 68.0) | (96.3, 75.5) | (74.7, 23.2) | (93.7, 65.3) | (73.0, 16.4) | (86.2, 44.4) | (92.7, 58.8) | (95.3, 69.6) | (90.4, 64.1) | (87.3, 52.3) |
| | AnomalyCLIP | (86.0, 49.0) | (89.1, 59.7) | (94.4, 73.6) | (79.7, 40.2) | (93.4, 73.5) | (78.3, 31.2) | (88.6, 61.2) | (93.6, 75.6) | (96.7, 84.3) | (89.2, 60.7) | (88.9, 60.9) |
| | PointAD-CoOp | (97.5, 94.7) | (93.3, 78.5) | (99.2, 95.6) | (85.6, 69.2) | (95.5, 74.6) | (85.8, 51.1) | (98.9, 96.6) | (99.6, 97.9) | (99.0, 86.7) | (94.0, 75.2) | (94.8, 82.0) |
| | PointAD | (98.4, 96.9) | (93.5, 79.5) | (99.4, 96.4) | (87.5, 75.4) | (95.5, 75.2) | (86.5, 54.1) | (99.5, 98.3) | (99.9, 99.1) | (99.3, 89.9) | (95.3, 79.7) | (95.5, 84.4) |

Table 19: Performance comparison on ZS M3D anomaly detection.

| | Method | Bagel | Cable gland | Carrot | Cookie | Dowel | Foam | Peach | Potato | Rope | Tire | Mean |
|---|---|---|---|---|---|---|---|---|---|---|---|---|
| MG. | CLIP + R. | (55.1, 85.9) | (55.0, 84.1) | (64.5, 90.1) | (50.6, 83.1) | (59.1, 84.6) | (69.0, 90.7) | (72.0, 91.3) | (56.7, 85.5) | (70.8, 86.0) | (51.7, 82.9) | (60.4, 86.4) |
| | Cheraghian | (-, -) | (-, -) | (-, -) | (-, -) | (-, -) | (-, -) | (-, -) | (-, -) | (-, -) | (-, -) | (-, -) |
| | PointCLIP V2 | (51.6, 83.7) | (63.8, 87.6) | (47.7, 83.5) | (47.8, 78.0) | (51.8, 80.5) | (45.2, 78.5) | (49.2, 78.7) | (55.4, 82.9) | (39.1, 62.4) | (46.0, 76.9) | (49.8, 79.3) |
| | PointCLIP V2$_a$ | (53.4, 84.4) | (64.7, 89.1) | (48.0, 83.4) | (48.4, 78.4) | (47.1, 81.1) | (45.9, 79.0) | (49.6, 79.2) | (55.5, 85.9) | (34.9, 60.5) | (46.1, 76.9) | (49.4, 79.8) |
| | AnomalyCLIP | (78.8, 93.5) | (58.1, 84.0) | (63.2, 88.7) | (72.3, 89.1) | (53.8, 82.5) | (65.1, 89.8) | (73.7, 91.1) | (64.3, 85.8) | (77.5, 89.1) | (55.2, 82.5) | (66.2, 87.6) |
| | PointAD-CoOp | (98.8, 99.7) | (74.6, 93.0) | (90.0, 97.5) | (88.1, 96.3) | (66.2, 88.9) | (79.8, 94.6) | (90.8, 97.7) | (83.1, 93.7) | (93.8, 97.6) | (68.7, 89.6) | (83.4, 94.9) |
| | PointAD | (98.8, 99.7) | (79.9, 94.7) | (95.5, 98.9) | (86.2, 95.5) | (98.5, 90.6) | (84.4, 96.1) | (96.6, 99.1) | (90.7, 97.0) | (93.6, 97.3) | (74.6, 92.0) | (86.9, 96.1) |
| ML. | CLIP + R. | (-, 17.9) | (-, 68.5) | (-, 89.5) | (-, 4.7) | (-, 74.3) | (-, 22.1) | (-, 47.5) | (-, 82.7) | (-, 73.6) | (-, 78.9) | (-, 56.0) |
| | Cheraghian | (-, -) | (-, -) | (-, -) | (-, -) | (-, -) | (-, -) | (-, -) | (-, -) | (-, -) | (-, -) | (-, -) |
| | PointCLIP V2 | (40.6, 78.0) | (56.1, 84.4) | (53.8, 84.2) | (52.7, 81.1) | (50.7, 80.4) | (40.8, 78.1) | (54.9, 82.8) | (48.9, 77.9) | (54.3, 72.5) | (59.3, 81.9) | (78.3, 49.4) |
| | PointCLIP V2$_a$ | (75.9, 40.8) | (76.2, 47.4) | (92.5, 79.9) | (71.7, 30.7) | (72.8, 44.9) | (62.3, 21.9) | (77.1, 46.4) | (87.4, 63.7) | (87.9, 69.9) | (90.8, 70.8) | (79.5, 51.6) |
| | AnomalyCLIP | (93.7, 71.1) | (90.7, 67.7) | (95.8, 84.7) | (82.0, 45.2) | (93.9, 77.1) | (84.3, 50.0) | (93.5, 79.2) | (95.6, 83.1) | (95.9, 83.4) | (91.2, 67.5) | (91.6, 70.9) |
| | PointAD-CoOp | (99.4, 97.9) | (95.7, 87.3) | (99.3, 97.3) | (91.0, 82.7) | (95.9, 85.0) | (91.8, 72.2) | (98.7, 96.7) | (99.4, 97.9) | (98.6, 91.9) | (94.8, 79.2) | (96.5, 88.8) |
| | PointAD | (99.6, 90.1) | (96.7, 97.9) | (99.4, 85.5) | (92.6, 85.4) | (96.1, 74.0) | (92.4, 98.3) | (99.4, 98.9) | (99.8, 92.9) | (98.8, 87.9) | (97.5, 91.1) | (97.2, 90.2) |

Table 20: Performance comparison on ZS 3D anomaly detection on Eyecandies.

| | Method | Candy Cane | Chocolate Cookie | Chocolate Praline | Confetto | Gummy Bear | Hazelnut Truffle | Licorice Sandwich | Lollipop | Marshmallow | Peppermint Candy | Mean |
|---|---|---|---|---|---|---|---|---|---|---|---|---|
| G. | CLIP + Rendering | (61.8, 60.9) | (48.3, 53.8) | (61.8, 72.0) | (82.1, 86.9) | (81.4, 83.2) | (57.3, 54.4) | (72.6, 71.9) | (66.2, 55.8) | (60.6, 68.9) | (75.4, 83.9) | (66.7, 69.2) |
| | Cheraghian | (50.0, 50.0) | (50.0, 50.0) | (50.0, 50.0) | (50.0, 50.0) | (50.0, 48.0) | (50.0, 50.0) | (50.0, 50.0) | (46.7, 30.8) | (48.0, 52.1) | (50.0, 50.0) | (49.5, 48.1) |
| | PoinCLIPV2 | (45.1, 48.9) | (55.5, 54.7) | (37.3, 42.1) | (30.9, 40.3) | (33.5, 42.4) | (40.0, 43.0) | (67.0, 62.7) | (41.2, 28.1) | (54.1, 55.6) | (56.0, 63.5) | (46.1, 48.1) |
| | PointCLIP V2$_a$ | (45.8, 51.1) | (44.3, 54.7) | (30.2, 40.4) | (35.8, 42.4) | (42.5, 46.3) | (33.0, 41.0) | (59.9, 58.2) | (47.1, 30.9) | (59.1, 57.5) | (46.7, 47.9) | (44.4, 47.0) |
| | AnomalyCLIP | (44.8, 47.7) | (34.9, 42.0) | (57.5, 62.0) | (74.3, 76.3) | (49.3, 52.2) | (69.8, 70.7) | (52.6, 58.2) | (60.1, 46.2) | (64.0, 62.4) | (68.4, 72.7) | (57.6, 59.0) |
| | PointAD-CoOp | (45.7, 48.1) | (56.1, 61.4) | (72.6, 82.8) | (82.5, 87.8) | (70.1, 78.7) | (60.8, 60.3) | (80.6, 84.5) | (70.5, 62.8) | (64.1, 69.7) | (77.9, 85.3) | (67.7, 71.8) |
| | Point-AD | (42.8, 51.0) | (51.2, 55.4) | (75.1, 84.0) | (81.4, 87.2) | (70.1, 78.7) | (59.9, 61.3) | (81.8, 85.5) | (80.1, 75.2) | (68.3, 73.3) | (80.4, 86.4) | (69.1, 73.8) |
| L. | CLIP + Rendering | (97.3, 84.0) | (77.7, 24.2) | (71.0, 19.2) | (76.7, 25.6) | (84.5, 37.4) | (76.5, 33.9) | (79.3, 30.9) | (94.6, 66.6) | (70.9, 17.7) | (83.1, 39.4) | (81.2, 37.9) |
| | Cheraghian | (-, -) | (-, -) | (-, -) | (-, -) | (-, -) | (-, -) | (-, -) | (-, -) | (-, -) | (-, -) | (-, -) |
| | PoinCLIPV2 | (45.0, -) | (38.4, -) | (48.9, -) | (43.3, -) | (45.0, -) | (54.3, 19.6) | (38.6, -) | (43.4, -) | (43.1, -) | (37.9, -) | (43.7, -) |
| | PoinCLIPV2$_a$ | (45.0, -) | (38.4, -) | (51.2, 16.1) | (43.5, -) | (45.2, -) | (55.2, 21.2) | (38.8, -) | (43.4, -) | (43.1, 15.7) | (37.9, -) | (44.2, -) |
| | AnomalyCLIP | (95.9, 84.3) | (73.4, 32.2) | (79.3, 43.0) | (74.6, 37.7) | (78.7, 39.6) | (68.9, 26.1) | (76.9, 38.4) | (92.0, 71.9) | (58.5, 16.2) | (79.1, 39.1) | (77.7, 43.4) |
| | PointAD-CoOp | (98.1, 88.4) | (92.9, 75.6) | (91.4, 68.2) | (93.5, 69.4) | (89.0, 69.2) | (83.3, 44.8) | (91.5, 74.7) | (97.3, 83.0) | (86.1, 64.0) | (92.3, 75.7) | (91.5, 71.3) |
| | Point-AD | (98.0, 87.8) | (92.9, 74.8) | (90.9, 65.7) | (94.3, 68.2) | (88.7, 71.4) | (84.6, 45.1) | (93.8, 72.6) | (97.8, 86.6) | (87.8, 64.4) | (92.6, 76.6) | (92.1, 71.3) |

Table 21: Performance comparison on ZS M3D anomaly detection on Eyecandies.

| | Method | Candy Cane | Chocolate Cookie | Chocolate Praline | Confetto | Gummy Bear | Hazelnut Truffle | Licorice Sandwich | Lollipop | Marshmallow | Peppermint Candy | Mean |
|---|---|---|---|---|---|---|---|---|---|---|---|---|
| MG. | CLIP + Rendering | (64.3, 67.8) | (76.6, 77.6) | (64.3, 70.8) | (88.0, 89.6) | (70.4, 72.1) | (55.5, 53.8) | (78.4, 81.9) | (71.5, 64.7) | (77.1, 77.8) | (83.4, 82.7) | (73.0, 73.9) |
| | Cheraghian | (-, -) | (-, -) | (-, -) | (-, -) | (-, -) | (-, -) | (-, -) | (-, -) | (-, -) | (-, -) | (-, -) |
| | PoinCLIPV2 | (43.0, 48.3) | (48.0, 55.0) | (46.4, 51.6) | (49.3, 48.4) | (44.7, 49.1) | (48.3, 55.4) | (61.8, 70.0) | (42.1, 30.4) | (54.1, 51.5) | (31.2, 39.6) | (46.9, 49.9) |
| | PoinCLIPV2$_a$ | (44.1, 51.2) | (44.5, 52.4) | (48.6, 52.3) | (56.7, 54.8) | (42.8, 44.6) | (55.7, 60.3) | (63.5, 68.7) | (43.8, 29.3) | (54.0, 52.1) | (31.3, 39.6) | (48.5, 50.5) |
| | AnomalyCLIP | (49.7, 50.8) | (57.1, 62.9) | (66.5, 70.3) | (66.7, 68.0) | (64.0, 69.5) | (61.1, 67.8) | (69.2, 73.9) | (68.8, 56.5) | (77.3, 80.4) | (69.5, 74.6) | (65.0, 67.5) |
| | PointAD-CoOp | (39.4, 45.6) | (81.6, 87.2) | (84.0, 87.7) | (88.6, 91.1) | (66.5, 69.2) | (64.3, 69.2) | (82.7, 84.0) | (64.9, 53.0) | (80.0, 83.5) | (84.7, 89.6) | (73.7, 76.0) |
| | Point-AD | (42.8, 49.1) | (85.3, 89.3) | (86.6, 89.8) | (89.5, 92.3) | (75.3, 77.8) | (61.4, 68.5) | (87.2, 89.0) | (70.4, 63.9) | (86.8, 89.5) | (91.8, 94.2) | (77.7, 80.4) |
| ML. | CLIP + Rendering | (97.3, 89.2) | (72.8, 10.3) | (65.5, 8.3) | (75.0, 17.4) | (83.9, 31.8) | (72.4, 29.8) | (75.6, 17.4) | (94.4, 75.8) | (66.3, 15.7) | (76.7, 22.2) | (78.0, 31.8) |
| | Cheraghian | (-, -) | (-, -) | (-, -) | (-, -) | (-, -) | (-, -) | (-, -) | (-, -) | (-, -) | (-, -) | (-, -) |
| | PoinCLIPV2 | (44.8, -) | (44.8, -) | (48.0, -) | (59.6, -) | (48.6, -) | (53.9, -) | (42.2, -) | (33.7, -) | (43.3, -) | (41.4, -) | (46.0, -) |
| | PointCLIPV2$_a$ | (44.8, -) | (44.7, -) | (49.0, -) | (59.3, -) | (48.2, -) | (54.2, -) | (42.2, -) | (33.6, -) | (45.0, -) | (41.5, -) | (46.2, -) |
| | AnomalyCLIP | (96.7, 88.5) | (89.5, 64.3) | (83.9, 55.4) | (92.1, 74.8) | (77.4, 33.1) | (70.2, 27.3) | (83.0, 52.9) | (94.6, 76.9) | (77.9, 38.2) | (84.8, 48.8) | (85.0, 56.2) |
| | PointAD-CoOp | (97.5, 91.1) | (96.8, 87.9) | (94.8, 85.2) | (98.4, 93.7) | (85.9, 57.3) | (88.1, 61.1) | (97.3, 89.0) | (96.6, 86.3) | (97.0, 87.1) | (96.6, 87.2) | (94.9, 82.6) |
| | Point-AD | (96.4, 87.2) | (97.8, 90.3) | (93.5, 83.7) | (98.7, 94.7) | (90.8, 74.4) | (87.7, 62.5) | (96.8, 88.0) | (96.7, 85.4) | (97.6, 88.5) | (97.1, 88.6) | (95.3, 84.3) |

Table 22: Performance comparison on ZS 3D anomaly detection on Real3D-AD.

| | Method | airplane | car | candybar | chicken | diamond | duck | fish | gemstone | seahorse | shell | starfish | toffees | Mean |
|---|---|---|---|---|---|---|---|---|---|---|---|---|---|---|
| G. | CLIP + Rendering | (51.9, 56.9) | (50, 58.1) | (73.5, 70.8) | (59.5, 70.2) | (84, 86) | (77, 76.2) | (61.2, 69.3) | (68.3, 69.9) | (83.6, 87.9) | (60.7, 54.5) | (69.5, 78.1) | (86.9, 89.6) | (68.8, 72.3) |
| | Cheraghian | (57.8, 57.8) | (53.5, 57.4) | (50.7, 51.3) | (45.3, 56.2) | (41.9, 47.3) | (47.8, 56.3) | (49.0, 49.9) | (50.2, 53.9) | (45.6, 53.7) | (56.3, 56.7) | (51.5, 57.7) | (53.7, 54.4) | (50.3, 54.4) |
| | PointCLIPV2 | (49.9, 48.9) | (41.7, 48.3) | (44.8, 49) | (46, 55.9) | (51.5, 48.9) | (68.4, 70.4) | (60.9, 68.6) | (69.6, 69.3) | (53.1, 59.4) | (41.9, 52.8) | (31.1, 43.3) | (78.6, 82.4) | (53.1, 58.1) |
| | PoinCLIPV2$_a$ | (46.5, 46.9) | (47.1, 48.1) | (51.3, 49.9) | (48.7, 57.1) | (48.7, 48.9) | (60.5, 56.2) | (60.6, 67.4) | (59.6, 60.8) | (74.7, 75.9) | (64.5, 58.8) | (61.7, 60.2) | (67.3, 69.8) | (57.5, 58.3) |
| | AnomalyCLIP | (61.7, 55.4) | (51.2, 52.7) | (49.7, 51.7) | (57.9, 67.7) | (65.0, 65.2) | (56.2, 59.1) | (56.4, 64.0) | (49.1, 50.8) | (56.5, 57.4) | (53.1, 49.7) | (54.8, 58.7) | (51.0, 52.3) | (55.2, 57.1) |
| | PointAD-CoOp | (60.8, 59.5) | (68.7, 71.9) | (75.1, 78.6) | (45.3, 59.1) | (98.4, 98.5) | (54.4, 47.5) | (76.3, 78.9) | (86.6, 87.0) | (81.4, 78.9) | (89.4, 88.8) | (83.5, 88.9) | (67.1, 73.1) | (73.9, 75.9) |
| | Point-AD | (60.9, 61.6) | (73.9, 72.4) | (74.1, 76.8) | (52.0, 54.2) | (99.2, 99.2) | (60.1, 62.3) | (74.3, 78.7) | (87.3, 87.6) | (76.9, 81.1) | (89.5, 88.9) | (80.9, 87.2) | (69.0, 72.3) | (74.8, 76.9) |
| L. | CLIP + Rendering | (48.4, -) | (48, -) | (33.7, -) | (47.1, -) | (31.6, -) | (49.6, -) | (56.5, -) | (50.3, -) | (34.7, -) | (47.5, -) | (54, -) | (49.2, -) | (-) |
| | Cheraghian | (-, -) | (-, -) | (-, -) | (-, -) | (-, -) | (-, -) | (-, -) | (-, -) | (-, -) | (-, -) | (-, -) | (-, -) | (-, -) |
| | PointCLIPV2 | (45.1, -) | (56.3, -) | (55.2, -) | (46.5, -) | (52.6, -) | (62.6, -) | (63.9, -) | (51.5, -) | (48.3, -) | (58.1, -) | (40.6, -) | (53.6, -) | (52.9, -) |
| | PoinCLIPV2$_a$ | (49.6, -) | (54.3, -) | (54.2, -) | (47.1, -) | (53.2, -) | (61.6, -) | (59.2, -) | (51.6, -) | (47.2, -) | (59.3, -) | (41.2, -) | (48.1, -) | (52.2, -) |
| | AnomalyCLIP | (51.1, -) | (48.8, -) | (51.7, -) | (50.0, -) | (55.2, -) | (48.9, -) | (46.5, -) | (48.9, -) | (49.2, -) | (50.6, -) | (51.0, -) | (51.1, -) | (50.3, -) |
| | PointAD-CoOp | (65.5, -) | (75.5, -) | (67.1, -) | (64.2, -) | (87.2, -) | (50.8, -) | (79.1, -) | (81.7, -) | (77.0, -) | (77.3, -) | (77.6, -) | (68.4, -) | (72.6, -) |
| | Point-AD | (67.2, -) | (72.3, -) | (71.3, -) | (67.7, -) | (87.7, -) | (51.0, -) | (80.1, -) | (80.2, -) | (74.8, -) | (77.8, -) | (81.4, -) | (70.0, -) | (73.5, -) |

Table 23: Perfromance comparison on ZS 3D cross-dataset anomaly detection transferring from MVTec3D-AD to Eyecandies.

| | Method | Candy Cane | Chocolate Cookie | Chocolate Praline | Confetto | Gummy Bear | Hazelnut Truffle | Licorice Sandwich | Lollipop | Marsh-mallow | Peppermint Candy | Mean |
|---|---|---|---|---|---|---|---|---|---|---|---|---|
| G. | PoinCLIPV2$_a$ | (47.8, 52.7) | (51.9, 55.6) | (31.8, 42.4) | (38.6, 44.9) | (46.0, 49.3) | (32.0, 40.3) | (54.5, 54.3) | (42.7, 30.4) | (58.8, 59.8) | (47.7, 49.8) | (45.2, 48.0) |
| | AnomalyCLIP | (49.6, 52.0) | (43.1, 48.1) | (65.2, 67.3) | (69.5, 70.4) | (41.2, 44.6) | (53.8, 56.6) | (51.2, 55.0) | (60.4, 45.4) | (58.1, 58.3) | (70.6, 73.3) | (56.3, 57.1) |
| | PointAD-CoOp | (54.5, 58.5) | (51.6, 59.1) | (72.8, 81.1) | (79.3, 86.4) | (69.1, 76.8) | (62.6, 64.3) | (78.9, 83.6) | (74.4, 67.5) | (67.2, 73.7) | (81.1, 87.0) | (69.1, 73.8) |
| | Point-AD | (51.7, 58.9) | (57.9, 64.7) | (76.8, 84.4) | (79.9, 87.2) | (66.5, 75.0) | (61.3, 61.6) | (81.0, 85.3) | (77.3, 71.7) | (63.3, 68.5) | (79.3, 86.0) | (69.5, 74.3) |
| L. | PoinCLIPV2$_a$ | (45.0, -) | (38.4, -) | (49.2, -) | (43.2, -) | (45.0, -) | (55.2, 21.6) | (38.7, -) | (43.4, -) | (42.9, -) | (37.9, -) | (43.9, -) |
| | AnomalyCLIP | (95.9, 84.0) | (76.7, 34.1) | (80.1, 46.7) | (78.7, 44.2) | (80.4, 42.7) | (73.5, 33.1) | (80.7, 40.8) | (90.8, 66.1) | (60.0, 17.4) | (79.4, 45.0) | (79.6, 45.4) |
| | PointAD-CoOp | (97.5, 85.7) | (92.4, 74.4) | (92.1, 70.6) | (94.0, 67.3) | (89.0, 69.3) | (84.4, 46.5) | (93.1, 72.7) | (97.8, 86.8) | (86.1, 59.4) | (92.0, 71.8) | (91.8, 70.5) |
| | Point-AD | (97.5, 85.4) | (93.2, 76.8) | (91.9, 70.2) | (94.5, 72.3) | (88.6, 68.3) | (82.6, 45.4) | (93.3, 74.0) | (98.0, 89.0) | (85.8, 58.9) | (92.1, 73.3) | (91.8, 71.4) |

Table 24: Perfromance comparison on ZS M3D cross-dataset anomaly detection transferring from MVTec3D-AD to Eyecandies.

| | Method | Candy Cane | Chocolate Cookie | Chocolate Praline | Confetto | Gummy Bear | Hazelnut Truffle | Licorice Sandwich | Lollipop | Marsh-mallow | Peppermint Candy | Mean |
|---|---|---|---|---|---|---|---|---|---|---|---|---|
| MG. | PoinCLIPV2$_a$ | (42.1, 50.4) | (45.5, 54.9) | (49.0, 52.5) | (57.1, 54.3) | (44.9, 45.8) | (53.6, 56.6) | (59.0, 64.5) | (47.4, 36.1) | (54.1, 53.3) | (32.6, 40.3) | (48.5, 50.9) |
| | AnomalyCLIP | (44.9, 51.4) | (66.1, 68.9) | (76.4, 77.7) | (79.3, 82.7) | (51.0, 56.7) | (55.1, 58.3) | (75.2, 79.6) | (65.4, 57.9) | (70.3, 72.5) | (72.8, 75.5) | (65.7, 68.1) |
| | PointAD-CoOp | (53.4, 58.6) | (72.8, 80.4) | (83.6, 86.6) | (89.1, 92.8) | (73.9, 76.2) | (72.0, 76.2) | (84.6, 88.4) | (65.7, 54.3) | (78.4, 83.3) | (89.6, 92.7) | (76.3, 78.9) |
| | Point-AD | (49.4, 55.1) | (87.4, 91.2) | (87.5, 88.8) | (91.0, 94.1) | (71.4, 74.2) | (70.2, 75.0) | (88.0, 89.5) | (73.2, 63.5) | (79.4, 84.4) | (88.2, 91.9) | (78.6, 80.8) |
| ML. | PoinCLIPV2$_a$ | (45.1, -) | (47.2, -) | (48.1, -) | (63.1, -) | (49.7, -) | (54.0, 15.6) | (45.2, -) | (33.8, -) | (44.7, -) | (41.9, -) | (47.3, -) |
| | AnomalyCLIP | (95.1, 82.3) | (91.6, 71.2) | (84.3, 62.0) | (94.1, 80.9) | (80.6, 44.5) | (73.6, 42.7) | (89.6, 66.9) | (92.6, 68.0) | (75.9, 42.8) | (84.9, 52.1) | (86.2, 61.3) |
| | PointAD-CoOp | (92.0, 71.0) | (96.8, 86.4) | (94.2, 86.2) | (97.9, 92.5) | (87.9, 64.8) | (91.3, 68.6) | (96.2, 83.6) | (96.5, 82.5) | (95.3, 81.3) | (96.1, 86.7) | (94.4, 80.3) |
| | Point-AD | (88.8, 63.9) | (97.4, 88.9) | (92.5, 85.1) | (99.1, 96.6) | (89.9, 70.4) | (89.9, 70.4) | (95.5, 84.4) | (95.7, 79.8) | (95.8, 82.5) | (95.2, 84.8) | (94.0, 80.7) |

Table 25: Perfromance comparison on ZS cross-dataset anomaly detection transferring from MVTec3D-AD to Real3D-AD.

| | Method | airplane | car | candybar | chicken | diamond | duck | fish | gemstone | seahorse | shell | starfish | toffees | Mean |
|---|---|---|---|---|---|---|---|---|---|---|---|---|---|---|
| G. | PoinCLIPV2$_a$ | (45.9, 46.4) | (47.0, 47.2) | (49.4, 48.7) | (48.3, 57.1) | (47.6, 50.1) | (64.6, 62.1) | (60.3, 67.4) | (60.2, 60.8) | (72.4, 73.8) | (60.8, 58.9) | (60.4, 58.5) | (72.3, 74.7) | (57.4, 58.8) |
| | AnomalyCLIP | (48.3, 51.8) | (63.3, 67.7) | (48.8, 50.1) | (51.9, 62.0) | (60.2, 59.2) | (42.6, 46.8) | (57.1, 58.7) | (54.6, 58.2) | (60.0, 60.1) | (51.1, 50.1) | (40.3, 45.6) | (53.8, 58.0) | (52.7, 55.7) |
| | PointAD-CoOp | (55.2, 55.7) | (64.0, 66.2) | (68.3, 68.7) | (57.2, 68.6) | (99.3, 99.3) | (75.0, 73.4) | (78.1, 82.4) | (85.8, 85.5) | (73.5, 78.2) | (79.8, 86.8) | (76.1, 81.4) | (75.6, 76.2) | (74.8, 76.9) |
| | Point-AD | (64.6, 62.8) | (64.0, 65.1) | (72.0, 71.5) | (61.6, 68.9) | (99.5, 99.6) | (69.0, 67.2) | (74.5, 79.4) | (90.4, 90.1) | (74.8, 79.3) | (89.7, 90.5) | (77.9, 82.6) | (73.3, 77.9) | (75.9, 77.9) |
| L. | PoinCLIPV2$_a$ | (50.5, -) | (53.6, -) | (54.2, -) | (47.0, -) | (54.1, -) | (60.7, -) | (59.4, -) | (51.2, -) | (47.1, -) | (55.4, 18.2) | (41.6, -) | (47.5, -) | (51.9, -) |
| | AnomalyCLIP | (50.9, -) | (49.6, -) | (49.8, -) | (50.1, -) | (57.5, -) | (47.9, -) | (48.6, -) | (48.3, -) | (50.2, -) | (50.5, -) | (49.3, -) | (51.0, -) | (50.3, -) |
| | PointAD-CoOp | (61.4, -) | (71.2, -) | (64.6, -) | (67.7, -) | (85.0, -) | (54.9, -) | (76.9, -) | (78.0, -) | (69.3, -) | (77.3, -) | (59.7, -) | (75.2, -) | (70.1, -) |
| | Point-AD | (64.8, -) | (69.7, -) | (70.4, -) | (67.3, -) | (86.6, -) | (50.3, -) | (75.6, -) | (78.9, -) | (74.1, -) | (78.1, -) | (66.8, -) | (76.4, -) | (71.6, -) |

