# OpenReview forum: "PointAD: Comprehending 3D Anomalies from Points and Pixels for Zero-shot 3D Anomaly Detection"
_NeurIPS.cc/2024/Conference — NeurIPS 2024 poster_

### Official Review · Reviewer_dthJ · 2024-07-13

**Soundness:** 3
**Presentation:** 3
**Contribution:** 3
**Rating:** 5
**Confidence:** 3

**Summary:**

The paper titled "PointAD: Comprehending 3D Anomalies from Points and Pixels for Zero-shot 3D Anomaly Detection" introduces PointAD, a novel framework designed for zero-shot 3D anomaly detection by leveraging both point clouds and RGB images. The approach builds on the strong generalization capabilities of CLIP and adapts it to 3D anomaly detection. The framework renders 3D anomalies into multiple 2D views and then projects these 2D representations back into 3D space. This hybrid representation learning allows for capturing generic anomaly semantics across various unseen 3D objects. PointAD also incorporates auxiliary point clouds to optimize learnable text prompts for better anomaly detection and segmentation.

**Strengths:**

1. The use of CLIP for 3D anomaly detection and the hybrid representation learning approach are novel contributions.
2. The theoretical foundations are robust, and the empirical validation is comprehensive.
3. The paper is clearly written, with well-structured sections and effective use of visual aids.
4. The approach addresses a critical gap in the field, with potential for significant impact on future research and practical applications.

**Weaknesses:**

1. The sensitivity of the method to the selection of rendering angles and the number of views could be explored further.
2. While the experimental results are compelling, additional validation on more diverse and challenging datasets would further strengthen the claims.
3. The discussion of potential limitations and future work could be expanded to provide a more comprehensive view of the method's applicability and areas for improvement.

**Questions:**

1. Could the authors provide more insights into the selection of rendering angles and the impact of different numbers of views on the detection performance?
2. Have the authors considered the robustness of the method under different environmental conditions, such as varying lighting and occlusions?
3. Could additional experiments on other industrial or medical datasets help to further validate the generalizability of the proposed method?


======== post rebuttal ==========
The authors' rebuttal solve most of my concerns, hence I raise my score to borderline accept.

**Limitations:**

The authors have addressed some limitations, including the fixed rendering angles and the potential for further optimization of view selection. Constructive suggestions for improvement include exploring the sensitivity to rendering parameters and validating the method under different environmental conditions.

---

> ### Author Rebuttal · Authors · 2024-08-07
>
> # Response to Reviewer dthJ
>
> We appreciate your recognition of our work. Your insightful suggestions help make our sensitivity analyses more comprehensive and convincing. In addition to the response to your valuable feedback, we conducted further explorations on rendering quality and input resolution (**Q1 and Q2 responses to Reviewer 1QVN**).
>
> **Q1:  Insights into the rendering angles and different numbers of views**
>
> PointAD interprets point clouds from their 2D renderings, where rendering angles and numbers collectively determine the amount of information derived. They have different emphases. For rendering angles, the importance lies in the discrepancy between adjacent angles, as this affects the information granularity that PointAD obtains from adjacent views. When the angle discrepancy is fixed, the number of renderings determines the coverage of 3D information in the resulting 2D renderings. To capture all point cloud information, especially abnormal points, it is crucial to ensure comprehensive coverage. Therefore, our approach in selecting rendering angles and the number of renderings is to guarantee that all points in point clouds are adequately represented.
>
> Based on this principle, we conducted experiments to assess the impact of the number of renderings on PointAD's detection performance, circularly rendering point clouds to ensure even coverage of all points. As shown in `Table 5 of the manuscript`, increasing the number of views allows PointAD to gather more detailed information from the 2D renderings, benefiting from smaller angle discrepancies, which improves detection and localization results. However, when the number of views increased from 9 to 11, we observed a performance decline in PointAD, with the I-AUROC for global multimodal detection dropping from 87.4\% to 86.4\%. This suggests that incorporating too many views could introduce redundant information, resulting in 2D renderings with extensive overlap and excessive local detail. This overemphasis on local information can impede global recognition.  Hence, the appropriate number of views benefits point understanding from informative views while mitigating the adverse effects of redundant local information.
>
> To further explore the impact of the absolute angle, we shift the rendering angles while keeping the angle discrepancy unchanged. The original adjacent angle discrepancy in our paper is $\frac{1}{5}\pi$. We divide this discrepancy into four parts and perform angle shifts of $\frac{1}{20}\pi$, $\frac{2}{20}\pi$, and $\frac{3}{20}\pi$ to test the impact of varying rendering angles. `Table 6` shows that PointAD maintains consistent performance across different rendering angles, demonstrating its robustness to variations in angles different from those used during training.
>
> **Q2: The robustness under varying lighting and occlusions.**
>
> Further exploring the robustness of PointAD under different conditions could enhance its generalized detection performance. We conducted ablation studies to test its sensitivity under different lighting conditions and with occluded point clouds below.
>
> To evaluate the impact of rendering lighting, we adjusted the lighting conditions to render point clouds, generating variant datasets with different lighting. We used both stronger and weaker lighting to render point clouds compared to the original dataset, covering a broad lighting range. We denote stronger and the strongest lighting as "+" and "++", and weaker and the weakest lighting as "-" and "--". Visualizations of the resulting samples are presented in `Figure 5`. The experiments were conducted on MVTec3D-AD, where we tested PointAD, trained on the original dataset, on these lighting variant datasets. `Table 7` shows that PointAD can still detect anomalies even with significant discrepancies in rendering lighting, suggesting that PointAD is not sensitive to variations in rendering lighting.
>
> Next, we evaluated detection performance with occluded point clouds. We occluded point clouds by removing those invisible from a specific rendering angle and then used the same rendering parameters to project the remaining points. During this process, we observed that abnormal regions might be occluded totally. Unlike class classification, where class semantics remain unchanged by point occlusions, removing these anomaly semantics would transform anomaly samples into normal points. Therefore, we selected rendering angles that allow visibility of part or all anomalous regions when the point cloud is an abnormal instance. The occluded point clouds are shown in `Figure 6`. We then used PointAD, trained on the original dataset, to test the resulting occluded dataset. `Table 9` shows that PointAD suffers from performance degradation when the point clouds are occluded. We attribute this to two aspects:
>
> 1. Despite this strategy, the occluded point clouds could still lose part of the anomaly semantics;
>
> 2. Occluded point clouds could create unexpected sinkholes on the surface, which may cause PointAD to identify these areas as hole anomalies incorrectly.
>
> **Q3:  Experiments on additional datasets**
>
> please refer to the bottom of **Response to Reviewer GgEU** for our response to **Q3** due to character limitation.
>
> **Q4: Discussion of potential limitations and future work.**
>
> We suggest potential improvements for PointAD as follows:
>
> 1. The current rendering process in PointAD does not involve optimization. Using fixed rendering angles for diverse objects might limit detection performance. Adapting rendering angles conditioned on specific objects could make 2D renderings more informative.
> 2. PointAD currently treats the 2D renderings as independent instances. Modeling the correlations between multiple 2D renderings based on view angles could enhance both 2D and 3D representations.
>
> If our responses address your concerns, we would appreciate it if you could consider raising your score to acknowledge our efforts in addressing your questions.

---

> ### Author Response · Authors · 2024-08-08
>
> Your support is greatly appreciated. We're more than happy to take any further questions if otherwise.

---

### Official Review · Reviewer_GgEU · 2024-07-14

**Soundness:** 3
**Presentation:** 3
**Contribution:** 3
**Rating:** 6
**Confidence:** 5

**Summary:**

The paper introduces PointAD, the first approach to explore the domain of zero-shot 3D anomaly detection, leveraging CLIP's strong generalization to identify anomalies in previously unseen objects. It offers a unified framework that integrates 3D point clouds with 2D renderings, employing hybrid representation learning to capture the semantics of anomalies. PointAD's key contributions are its pioneering exploration of the ZS 3D anomaly detection domain, its ability to seamlessly integrate multimodal data like RGB information for enhanced detection, and its superior performance over existing methods. The robustness of the framework is confirmed through extensive experiments.

**Strengths:**

Originality: The paper introduces PointAD, a novel method in 3D anomaly detection that leverages CLIP for 3D analysis, uniquely combining point clouds and pixel data. It expands the application of vision-language models into new domains, showcasing versatility in 3D point cloud analysis.
Soundness: The paper exhibits methodological soundness through its rigorous experimental setup, including the use of diverse datasets and a thorough ablation study that substantiates the design decisions and effectiveness of the proposed PointAD framework. The state-of-the-art experiments performance confirms the model's outperforming ability.
Clarity: The paper stands out for its clarity, guiding readers smoothly from the problem statement to the final results. It skillfully explains complex concepts in an accessible way, ensuring that a wider audience can follow along. The paper also benefits from helpful visual aids that clearly demonstrate the model's effectiveness.

**Weaknesses:**

1.	There’s some zero-shot anomaly detection methods with clip on 2D images, so this paper should compare with them, especially, from the view of techniques’ differences, not only the used data differences. For example, those zero-shot methods (WinCLIP, VAND and AnomalyCLIP) mentioned in [R1] and [R2].
[R1] Cao, Y., Xu, X., Zhang, J., Cheng, Y., Huang, X., Pang, G., & Shen, W. (2024). A survey on visual anomaly detection: Challenge, approach, and prospect. arxiv preprint arxiv:2401.16402.
[R2] Li, X., Huang, Z., Xue, F., & Zhou, Y. (2024). Musc: Zero-shot industrial anomaly classification and segmentation with mutual scoring of the unlabeled images. arxiv preprint arxiv:2401.16753.

2.	I also wonder if the new methods of 3D anomaly detection can also be used to deal with this zero-shot task. For example, [R3] and [R4].
[R3] Zuo, Z., Dong, J., Wu, Y., Qu, Y., & Wu, Z. (2024). CLIP3D-AD: Extending CLIP for 3D Few-Shot Anomaly Detection with Multi-View Images Generation. arxiv preprint arxiv:2406.18941.
[R4] Li, W., Xu, X., Gu, Y., Zheng, B., Gao, S., & Wu, Y. (2024). Towards Scalable 3D Anomaly Detection and Localization: A Benchmark via 3D Anomaly Synthesis and A Self-Supervised Learning Network. In Proceedings of the IEEE/CVF Conference on Computer Vision and Pattern Recognition (pp. 22207-22216).
3.	More importantly, there’s some very similar studies with this work and task. Authors should mention and compare them within this study. For example, [R5].
[R5] Li, Y., Goodge, A., Liu, F., & Foo, C. S. (2024). Promptad: Zero-shot anomaly detection using text prompts. In Proceedings of the IEEE/CVF Winter Conference on Applications of Computer Vision (pp. 1093-1102).
[R6] X. Chen, J. Zhang, G. Tian, H. He, W. Zhang, Y. Wang, C. Wang,Y. Wu, and Y. Liu, “Clip-ad: A language-guided staged dual-
path model for zero-shot anomaly detection,” arXiv preprint
arXiv:2311.00453, 2023.
[R7] Wang, C., Zhu, H., Peng, J., Wang, Y., Yi, R., Wu, Y., ... & Zhang, J. (2024). M3DM-NR: RGB-3D Noisy-Resistant Industrial Anomaly Detection via Multimodal Denoising. arxiv preprint arxiv:2406.02263.
4.	Computational Efficiency: Although the authors have commendably addressed computational challenges, the overall computational cost associated with rendering multi-view images and processing them through the vision encoder is still substantial. Since this field is still developing with few comparable studies, it is recommended that the authors provide a broader comparison about the computational cost with existing anomaly detection techniques, including those like Cheraghian, CPFM and 3DSR that are not strictly within the zero-shot category. This expanded comparison will help to show that the computational demands of PointAD are reasonable and suitable for practical use cases.
5.	Clarity and Completeness of Presentation: The paper's writing is generally clear and free of errors, yet minor aspects of the presentation could be polished for better reader comprehension. For instance, the figure overlays, specifically in Figure 1, currently only elucidate part (a), leaving explanations for parts (b) and (c) to be found in the main text. This separation can disrupt the reader's flow and understanding. Enhancing the figure overlays to include all relevant parts and ensuring that each component of the figure is self-explanatory will make the paper more accessible and its findings easier to grasp.

**Questions:**

1.	Please answer the main differences with these mentioned methods in Weaknesses and compare with part of them on experiments.
2.	Given the concerns raised in the Weaknesses section 1, does PointAD still hold the merit of computational efficiency when compared to other methods such as AnomalyCLIP and Cheraghian?
3.	Could you elaborate on the multi-view rendering process of point clouds (as described from line 134 to line 140), particularly how the color of the rendered images is determined? Furthermore, at line 254, it's stated that additional RGB information is utilized only in the test dataset. Given the description at line 564, which refers to 2D RGB information, how is this 2D data integrated with the rendered multi-view images? Are there any strategies which are employed to bridge the gap between the rendered training data and the RGB information used in testing?

**Limitations:**

The authors have discussed the limitations of their model, and there are no potential negative societal impacts associated with their work.

---

> ### Author Rebuttal · Authors · 2024-08-07
>
> # Response to Reviewer GgEU
>
> Thanks for acknowledging our work. Your insightful comments help highlight our technological contributions in comparison to concurrent works and further promote comprehensive evaluations.
>
> Due to the character limit, please refer to the bottom of **General Response** for our reply to **Q1**.
>
> **Q2: Please compare with part of them on experiments.**
>
> We have included these simplified illustrations in the revised version. For experimental comparisons, since the source codes for PromptAD, CLIP3D-AD, IMRNet, and M3DM-NR are not available, we compared PointAD's performance with WinCLIP and VAND (AnomalyCLIP has been incorporated in the current submission). Note that VAND and CLIP-AD are similar methods, and we selected VAND for comparison due to time constraints. `Table 4` shows that PointAD significantly outperforms these baselines across all datasets in the one-vs-rest setting. The consistent superiority across diverse baselines demonstrates the effectiveness of PointAD. Due to the page limitation, we will provide results for the cross-dataset setting in the revised version.
>
> **Q3: Does PointAD still hold the merit of computational efficiency when compared to other methods such as AnomalyCLIP and Cheraghian?**
>
> Thanks for pointing out your concern. We have supplemented the computation overhead of WinCLIP, VAND, and AnomalyCLIP. In addition, as you recommended in Weakness Section 2, we report the computation consumption of unsupervised methods such as CPFM and 3DSR. From `Table 5`, we can observe that PointAD consumes medium computation overhead to achieve the best results. A lightweight version of PointAD is a worthwhile direction to explore for future work.
>
> **Q4: This separation can disrupt the reader's flow and understanding.**
>
> Thanks for your detailed feedback. We agree that the incomplete figure captions could confuse readers. We will supplement the necessary captions to all figures in the manuscript to enhance clarity.
>
>
> **Q5: Could you elaborate on the multi-view rendering process of point clouds (as described from line 134 to line 140), particularly how the color of the rendered images is determined ?**
>
> Given the point clouds and their corresponding point-level labels, we first resize both the point clouds and labels to 336 × 336. To represent the 3D point clouds in multiple 2D renderings, we use the Open3D library to render the point clouds from various views in a circular manner to preserve 3D information. We use gray ([0.7, 0.7, 0.7]) as the rendering color for all objects in our experiments for two main reasons:
> 1. Prominent colors might introduce a correlation between anomaly semantics and color information, potentially reducing PointAD's generalization capacity. Using gray helps mitigate this color-induced bias in detecting 3D anomalies.
>
> 2. Gray effectively represents 3D anomalies in 2D renderings, facilitating PointAD’s ability to capture the corresponding anomaly semantics.
>
> PointAD requires corresponding 2D labels to explicitly align 3D and 2D anomalies. Therefore, we color the 3D point clouds according to the point labels, with normal regions marked as 0 (black, including the background) and abnormal regions as 1 (white). We then render these from the same angles to obtain the corresponding 2D pixel labels.
>
> **Q6: Given the description at line 564, which refers to 2D RGB information, how is this 2D data integrated with the rendered multi-view images? Are there any strategies which are employed to bridge the gap between the rendered training data and the RGB information used in testing?**
>
> Thank you for raising your concern. Since 2D RGB data includes the RGB information of all points, we directly feed this 2D data into PointAD to obtain both the RGB anomaly scores (global) and the anomaly score maps (local). As shown in `Figure 4`, the RGB information can be considered as the additional 2D rendering that includes RGB information of all points. Unlike 2D RGB information, 2D renderings from different views represent only partial information. We need to aggregate these renderings to generate the point anomaly scores and point anomaly score maps. For fusion, we combine the RGB point anomaly score and the anomaly score maps using a weighted average.
>
> We do not use any additional strategies for fusing RGB and point clouds. The effective fusion is due to the explicit alignment of 3D and 2D spaces, which allows PointAD to learn from both point and pixel anomalies collaboratively. This approach captures generic abnormal patterns that are independent of object class semantics and colors, enabling PointAD to generalize well to RGB information during testing. It is worthwhile to explore a fusion mechanism to further bridge this gap. One potential approach could be to introduce 2D rendering information as a condition for representation learning, which might improve PointAD’s generalization to unseen class semantics.
>
> We appreciate your acknowledgment of our work. If our responses satisfy you, we would be grateful if you could consider improving your scores.
>
> ---
> # Due to character limitation, we continue the response to **Q3** for **Reviewer dthJ** here.
>
> **Q3:  Experiments on additional datasets**
>
> Since 3D anomaly detection is an emerging field, publicly available 3D datasets are limited. Fortunately, we have found a newly available industrial dataset, Anomaly-ShapeNet. However, we fail to find a point cloud dataset related to medical diagnostics and plan to incorporate such data when it becomes available.
>
> Anomaly-ShapeNet is a challenging dataset, comprising 1600 point clouds from 40 classes with 6 anomaly types, without RGB information. We conducted a one-vs-rest setting, using ashtray0, bottle, and cup0 separately to train PointAD. The averaged results on the remaining classes are reported in `Table 9`. The results indicate that PointAD consistently achieves superior performance by a significant margin compared to other baselines.

---

> > ### Comment · Reviewer_GgEU · 2024-08-12
> >
> > Thanks for the authors' response to my concerns. However, I found that you just compared yours with the new relevant papers I mentioned regarding technique differences. Ok, here I'll give two simple questions:
> >
> > 1) As you mentioned that "PointAD transfers the 2D generalization of CLIP to 3D point clouds and has the detection capacity for zero-shot 3D and M3D anomalies.". So, I wonder what your main innovations are from this study.
> >
> > 2) This study also used the MVTec-3D AD dataset, and this dataset has been cited with more than 40 references now. Why do you not compare yours with those new methods on this dataset? e.g., [1] with available codes; and others.
> >
> >
> > [1] Wang, Y., Peng, J., Zhang, J., Yi, R., Wang, Y., & Wang, C. (2023). Multimodal industrial anomaly detection via hybrid fusion. In Proceedings of the IEEE/CVF Conference on Computer Vision and Pattern Recognition (pp. 8032-8041).

---

> > > ### Author Response · Authors · 2024-08-13
> > > **Response for Q1**
> > >
> > > Thank you for pointing out your concerns. Please allow me to brief review the previous rebuttal. We have supplemented the performance comparison with WinCLIP and VAND in the previous response to **Q2**. As for the remaining methods, we do not compare them (i.e., PromptAD, CLIP3D-AD, IMRNet, and M3DM-NR) because they do not release their codes.
> > >
> > > Next, we hope our responses could further help address your concerns.
> > >
> > > **Q1: The main innovations from this study**
> > >
> > > The innovation of PointAD can be summarized in three key aspects: field contribution, main technological contribution, and applicability contribution.
> > >
> > >
> > > **Field contribution:** Zero-shot 3D anomaly detection is a promising task, to which the conventional unsupervised methods are not visible, when the target 3D samples are unavailable due to privacy concerns or the absence of new products such as those generated by popular 3D generation fields. To the best of our knowledge, PointAD is the first to explore the challenging yet significant zero-shot task of detecting and localizing point anomalies across diverse objects and datasets. Our paper paves the way for future research in this unexplored field.
> > >
> > >
> > > **Main technological contribution:** While CLIP demonstrates strong generalization capabilities for downstream tasks, its zero-shot performance for point anomaly detection and localization is limited (as shown in Tables 1 and 2 of the submitted manuscript). PointAD addresses this by proposing hybrid representation learning with tiny learnable parameters (only two text prompts), enabling CLIP to achieve accurate zero-shot 3D detection and zero-shot M3D detection.
> > >
> > > The collaborative optimization innovatively captures generalized point anomaly semantics from both 3D and 2D perspectives. This enables PointAD to understand 3D anomalies through both point space and pixel space. Benefiting from it, PointAD could achieve zero-shot M3D in a plug-and-play manner, without any additional training.
> > >
> > > **Applicability contribution:** PointAD transfers the extensive computation of point-level anomaly maps to a smaller feature space, significantly reducing memory consumption. This optimization improves PointAD's computational efficiency, allowing it to run on a single NVIDIA RTX 3090 24GB GPU, which enhances its applicability to real-world scenarios.

---

> ### Author Response · Authors · 2024-08-13
> **Response for Q2**
>
> **Q2: Compare PointAD with those new methods on MVTec-3D AD**
>
>  We find SOTA unsupervised methods (i.e., M3DM [1], BTF [2], and SDM [3]), which provide the codes. **We are pleased to compare these unsupervised methods as the upper boundary of performance.** Therefore, we follow their original settings, training category-specific models to report their results. Additionally, we also include them in the comparison of computation overhead. **These results will be incorporated into the revised version.**
>
> |Methods|Point detection (Global)|Point detection (Local)|Multimodal detection (Global)|Multimodal detection (Local)|
> |:---:|:---:|:---:|:---:|:---:|
> |BTF (unsupevised)|76.3, 91.8|97.6, 92.3|89.8, 96.7|99.5, 97.1|
> |SDM (unsupevised)|96.7, 90.9|97.0, 91.8|92.1, 97.6|93.3, 98.1|
> |M3DM (unsupevised)|85.6, 93.4|92.8, 91.6|93.2, 92.7|98.4, 95.1|
> |CPFM(unsupervised)|94.9, 98.7|97.6, 92.5|-,-|-,-|
> |3DSR(unsupervised)|95.1, 94.3 |93.8, 91.2|97.3, 98.6|99.3, 97.4|
> |WinCLIP (zero-shot)|45.2, 77.9|85.8, 59.4|38.7, 74.1|87.5, 64.2|
> |VAND (zero-shot)|55.0, 83.4|86.4, -|68.6, 89.2|85.2, -|
> |Ours (zero-shot)|82.0, 92.4|95.5, 84.4|86.9, 96.1|97.2, 90.2|
>
> From Table 1, we can observe that PointAD even outperforms the unsupervised method on certain metrics. Specifically, PointAD outperforms BTF in global point detection, achieving 82.0 I-AUROC and 92.4 AP, compared to BTF's 76.3 I-AUROC and 91.8 AP. Besides, PointAD obtains higher P-AUROC for local point detection compared to M3DM, with scores of 95.5 vs. 92.8. It also surpasses SDM with a P-AUROC of 97.2 vs. 93.3 in the local multimodal detection. Although there is still room for improvement compared to these unsupervised methods, these results demonstrate the superiority of PointAD.
>
> |Methods|inference time (s)|FPS|GPU (MB)|Point detection (Global)|Point detection (Local)|Multimodal detection (Global)|Multimodal detection (Local)|
> |:---:|:---:|:---:|:---:|:---:|:---:|:---:|:---:|
> |BTF (unsupevised)|0.18|5.56|1934|76.3, 91.8|97.6, 92.3|89.8, 96.7|99.5, 97.1|
> |SDM (unsupevised)|0.14|7.14|2716|96.7, 90.9|97.0, 91.8|92.1, 97.6|93.3, 98.1|
> |M3DM (unsupevised)|2.86|0.35|7494|85.6, 93.4|92.8, 91.6|93.2, 92.7|98.4, 95.1|
> |CPFM(unsupervised)|0.22 |4.55 |1379 |94.9, 98.7|97.6, 92.5|-,-|-,-|
> |3DSR(unsupervised)|0.09 |11.11 |3067| 95.1, 94.3 |93.8, 91.2|97.3, 98.6|99.3, 97.4|
> |WinCLIP (zero-shot)|0.29|3.45|3914|45.2, 77.9|85.8, 59.4|38.7, 74.1|87.5, 64.2|
> |PointCLIP V2 (zero-shot)|1.52|0.66|9747|78.3, 49.4|87.4, 52.3|49.8, 79.3|51.2, 80.1|
> |Cheraghian (zero-shot)|0.35 |2.86 |4847 | 53.6,81.7 | 88.2,57.0 |-,- | -,-|
> |Ours (zero-shot)|0.40 |2.52 |4275|82.0, 92.4|95.5, 84.4|86.9, 96.1|97.2, 90.2|
>
> Table 2 demonstrates that PointAD strikes a balance between computational overhead and performance. However, further reducing memory consumption remains a potential area for improvement.
>
> If you have any further concerns or questions, please let us know. We are more than happy to address them.
>
> **References:**
>
> [1] Wang, Y., Peng, J., Zhang, J., Yi, R., Wang, Y., \& Wang, C. (2023). Multimodal industrial anomaly detection via hybrid fusion. In Proceedings of the IEEE/CVF Conference on Computer Vision and Pattern Recognition (pp. 8032-8041).
>
> [2] Horwitz, E., \& Hoshen, Y. (2023). Back to the feature: classical 3d features are (almost) all you need for 3d anomaly detection. In Proceedings of the IEEE/CVF Conference on Computer Vision and Pattern Recognition (pp. 2968-2977).
>
> [3] Chu, Y. M., Liu, C., Hsieh, T. I., Chen, H. T., \& Liu, T. L. (2023, July). Shape-guided dual-memory learning for 3D anomaly detection. In Proceedings of the 40th International Conference on Machine Learning (pp. 6185-6194).

---

### Official Review · Reviewer_1QVN · 2024-07-20

**Soundness:** 3
**Presentation:** 3
**Contribution:** 3
**Rating:** 7
**Confidence:** 3

**Summary:**

The paper propose a unified framework, PointAD,to detect 3D anomalies in a ZS manner. Hybrid representation learning is proposed to incorporate the generic normality and abnormality semantics into PointAD. PointAD can incorporate 2D RGB information in a plug-and-play manner for testing, which can perform ZS M3D anomaly detection directly. Experiments results demonstrate the superiority of the model in detecting and segmenting 3D anomalies.

**Strengths:**

1. PointAD introduces a novel method for Zero-Shot 3D anomaly detection by integrating the strong generalization capabilities of CLIP to recognize 3D anomalies from unseen objects. This approach is first of its kind in ZS 3D anomaly detection.
2. The paper describes a unified framework that comprehends 3D anomalies from both points and pixels, allowing for a comprehensive understanding of anomalies by rendering 3D objects into 2D renderings and projecting them back to 3D space. It proposes hybrid representation learning to optimize learnable text prompts from both 3D and 2D data, enhancing the model's ability to generalize across diverse unseen objects.
3. Extensive experiments demonstrate the superiority of PointAD over existing state-of-the-art methods, especially in ZS settings where traditional models struggle.

**Weaknesses:**

I do not notice any obvious weaknesses in this paper, but I still have a few questions that I hope the authors can answer.
1. The methodology, while innovative, is complex due to the need to render 3D objects into multiple 2D views and then re-project them, the performance of the model may heavily depends on the quality of the 2D renderings of the 3D objects, Therefore, if the renderings are of poor quality, could the model's performance degrade?
2. The experiments in the paper utilize high-resolution point cloud scans. Given that the point clouds obtained in practical applications may not always be high resolution, how effective is this method in detecting anomalies in low-resolution, sparse scans?
3. In the appendix's failure case section, the detection of the tire failed on the point score map. Thus, if a tire, or any object, has noise, how significantly does it affect the recognition on the point score map? I suspect that the possible reason for this failure could be the algorithm's sensitivity to point density and spatial distribution. When anomalies are located in areas where point density or distribution are inconsistent or significantly different from the training data, can this model still generalize well to new, unseen anomalies?

**Questions:**

See weaknesses.

**Limitations:**

The authors adequately addressd the limitations. Societal impact is not discussed, but I believe it's not needed.

---

> ### Author Rebuttal · Authors · 2024-08-07
>
> # Response to Reviewer 1QVN
>
> Thank you very much for your acknowledgment and insightful comments. Your comments inspire us to explore the robustness of PointAD from the rendering process and input under different conditions.
>
> **Q1: If the renderings are of poor quality, could the model's performance degrade?**
>
> Thank you for pointing out your concerns. PointAD interprets point clouds through their corresponding 2D renderings, and the quality of these renderings impacts the information that PointAD can extract from the original point clouds. In our manuscript, we used the Open3D Library to render the point clouds, but it does not provide an API for controlling rendering quality. To simulate varying rendering quality, we applied Gaussian blur with different extents ($\sigma$) to the 2D renderings. Sample visualizations are included in `Figure 1`. Specifically, we conducted experiments on MVTec3D-AD using different blur sigma values (i.e., $\{1, 5, 9\}$). `Table 1` shows that the detection performance of PointAD diminishes as rendering quality decreases (with increasing sigma). However, the degradation is acceptable even when the renderings are heavily blurred (sigma equals 9). In such cases, PointAD still outperforms baselines that use high-quality renderings.
>
> **Q2: How effective is this method in detecting anomalies in low-resolution, sparse scans?**
>
> We appreciate your detailed feedback, which has been invaluable in refining our work. To create low-resolution point clouds, we downsample the entire high-resolution point clouds using Farthest Point Sampling (FPS) with various sampling ratios. This strategy allows us to generate corresponding low-resolution datasets for training and evaluating PointAD. Visualizations of these datasets are provided in `Figure 2`. We train PointAD using the resulting low-resolution samples and test PointAD on the same resolution.
>
> `Table 2` demonstrates that PointAD maintains a strong detection capacity for low-resolution point clouds when the downsampling ratios are 20\%, 30\%, 50\%, and 70\%. Even at 20\% resolution, PointAD still achieves state-of-the-art performance. This indicates that PointAD is generally applicable to point clouds with various resolutions.
>
> **Q3: How significantly does the noise affect the recognition on the point score map?**
>
> Thanks for your insightful comments. We agree that failure detection could arise from unusual point density and distribution. To demonstrate the effect of point density, we randomly select normal regions of point clouds and subsequently increase or decrease the density of these regions through upsampling and downsampling. We provide a qualitative analysis in `Figure 3`. The visualization shows that PointAD effectively resists noise at reasonable levels. However, when noise levels are extremely low or high, the corresponding regions become excessively sparse or dense. This causes normal regions to appear similar to hole anomalies or squeezed anomalies, leading PointAD to classify these noisy areas as anomalies.
>
> **Q4: When anomalies are located in areas where point density or distribution are inconsistent or significantly different from the training data, can this model still generalize well to new, unseen anomalies?**
>
> To investigate the impact of point density differences between the training and test datasets, we train PointAD using high-density point clouds (original dataset) and then test it on the low-density versions of the datasets, downsampled as described in **Q2**. `Table 3` shows that PointAD can still detect anomalies even when retaining 50\% of the points from the original point clouds. However, when more points are removed (30\% and 20\% sample ratio), PointAD experiences an obvious performance degradation. We attribute the misdetection to the overly sparse point clouds forming holes. Nevertheless, PointAD can still detect anomalies even with a significant gap in point density between the training and test domains (e.g., 100\% vs. 20\%). This demonstrates PointAD's ability to generalize across different point densities.
>
> **In summary, the experimental results demonstrate PointAD's robustness to diverse rendering conditions. We attribute this robustness and generalization to PointAD's efficient hybrid representation learning and CLIP's powerful feature extraction. If you are satisfied with our responses, we would greatly appreciate it if you could consider raising your score. Your support is very important to us.**

---

> > ### Comment · Reviewer_1QVN · 2024-08-12
> >
> > The author's responses have addressed my concerns. I'm willing to raise my score to accept

---

> > > ### Author Response · Authors · 2024-08-13
> > >
> > > We're glad to hear that your concerns have been addressed. If you have any further questions, please don’t hesitate to let us know. We are more than willing to assist with any remaining issues.

---

### Author Rebuttal · Authors · 2024-08-07

# General Response
Dear Reviewers and ACs,

We very much appreciate the insightful and detailed review. We are delighted to hear the encouraging comments from all of the reviewers, including "`novel approach`" (Reviewer **1QVN** and **dthJ**), "`methodological soundness`" (Reviewer **GgEU**), "`stands out for its clarity and easy to follow`" (Reviewer **GgEU** and **dthJ**), and "`robust theoretical foundation and comprehensive empirical validation`" (Reviewer **dthJ**), "`potential for significant impact on future research and practical applications`" (Reviewer **dthJ**), and "`not any obvious weaknesses`" (Reviewer **1QVN**).  While we responded to each of the reviewer comments individually, we also provide a brief summary of the main contents of the rebuttal in response to the reviews:

- In response to the feedback from Reviewer **1QVN**, we have supplemented a deep analysis of the impact of rendering quality, input solution, and point density and distribution.
- As recommended by Reviewer **GgEU**, we have supplemented more baselines to further demonstrate the superiority of our method in performance and computation efficiency.
- To address the concerns raised by Reviewers **dthJ**, we have added more sensitivity analysis (illustrations and experiments) including rendering angles, rendering number, and robustness under different environments. Also, an additional industrial dataset is incorporated into the evaluation.
- We have addressed the feedback from Reviewers **GgEU** by incorporating important references that were previously missing. And, more potential limitations and future work are discussed to respond to Reviewers **dthJ**.

**We put Figures and Tables in the submitted PDF attached to the General Response due to the character limitation.**

- For Reviewer **1QVN**, we provide visualizations on 2D renderings with different quality in `Figure 1`, low-resolution point clouds in `Figure 2`, and noisy point clouds in `Figure 3`. Also, we provide the corresponding quantitive experimental results in `Tables 1, 2, and 3`.
- For Reviewer **GgEU**, we provide the schematic of multimodality 3D inference in `Figure 4`. `Tables 4 and 5` are supplemented to perform a more comprehensive performance comparison and analysis of computation overhead.
- For Reviewer **dthJ**, `Tables 6, 7, and 8` are added to respond to the concerns about rendering angle, rendering lighting, and point occlusions. Another additional dataset is tested in `Table 9`. `Figures 5 and 6` present the visualizations of point clouds under varying rendering lighting and occluded points.

For other questions raised by the reviewers, please see our response to individual questions and concerns below each review.

---
# Due to the character limitation, we respond to Q1 for Reviewer GgEU here.

**Q1: Please answer the main differences with these mentioned methods in Weaknesses.**

We thank the reviewer for pointing us toward these concurrent works [1][2]. Since our paper explores an emerging field, i.e., 3D zero-shot anomaly detection, PointAD bridges the gap between zero-shot anomaly detection (previously focused on 2D objects) and 3D anomaly detection. We agree that comparing these approaches could shed light on PointAD. Analysis of the differences between PointAD and these methods are as follows.

**PointAD (3D zero-shot) vs. 2D zero-shot**
First, the scope is different. PointAD transfers the 2D generalization of CLIP to 3D point clouds and has the detection capacity for zero-shot 3D and M3D anomalies. These works aim to capture planar anomalies rather than spatial anomalies.

- **PointAD vs. (WinCLIP, VAND, and CLIP-AD):** They require extensive human-crafted text prompts to provide representative textual embedding for identifying normality and abnormality. Instead, PointAD can perform zero-shot 3D and M3D anomaly detection using only two learnable generic textual prompts with the help of an auxiliary dataset.

- **PointAD vs. AnomalyCLIP:** AnomalyCLIP limits its prompt learning to the pixel level. In contrast, PointAD proposes hybrid representation learning to capture point and pixel abnormality simultaneously. This mutual collaboration enhances the understanding of both point and pixel anomalies, enabling PointAD to achieve superior performance.

- **PointAD vs. PromptAD:** PromptAD focuses on effectively fusing high-level embeddings to enhance zero-shot detection performance. In contrast, PointAD adapts CLIP from a lower level, prioritizing learning the text prompt to refine the textual embedding.

**PointAD (3D zero-shot) vs. 3D unsupervised**
The differences in settings lead to discrepancies in model design. 3D unsupervised methods fit models to specific categories, limiting their ability to detect unseen objects. In contrast, the zero-shot paradigm learns general decision boundaries, improving the detection of unseen objects.

- **PointAD vs. CLIP3D-AD:** CLIP3D-AD requires extensive human-crafted prompts to produce textual embeddings and learns to capture anomaly semantics only from 2D renderings (pixel anomalies). In contrast, PointAD requires just two learnable text prompts to learn category-agnostic textual embeddings for detecting anomalies in unseen objects. More importantly, PointAD understands 3D anomalies by collaboratively learning from both 3D and 2D perspectives, making it more aware of the spatial relationships of abnormal points.

- **PointAD vs. (IMRNet and M3DM-NR):** They store and model category-wise normal distributions using reconstruction or memory mechanisms. Instead, PointAD leverages CLIP's generalization to model the generic anomaly semantics between textual and visual embeddings. This allows PointAD to achieve both 3D anomaly and multimodal detection.

> **Reference:**
> 1. A survey on visual anomaly detection: Challenge, approach, and prospect.
> 2. Musc: Zero-shot industrial anomaly classification and segmentation with mutual scoring of the unlabeled images.

---

### Comment · Area_Chair_Hwnz · 2024-08-07

Hi reviewers,

Thanks a bunch for all your hard work as reviewers for NeurIPS 2024.

The discussion period between reviewers and authors has started. Make sure to check out the authors' responses and ask any questions you have to help clarify things by 8.13.

Best,
AC

---

> ### Comment · Area_Chair_Hwnz · 2024-08-12
>
> Dear reviewers,
>
> As the reviewer-author discussion period is about to end by 8.13, please take a look at other reviewers' reviews and authors' rebuttal at your earliest convenience. It would be great if you could ask authors for more clarification or explanation if some of your concerns are not addressed by the rebuttal.
>
> Thanks,
>
> AC

---

### Decision · Program_Chairs · 2024-09-25

**Decision:**

Accept (poster)

**Comment:**

The submissions got three positive recommendations. The reviewers liked this interesting idea for 3D anomaly detection, but also had some concerns on novelty, limitations, and effectiveness on more datasets. The authors addressed these concerns in the rebuttal and the comments during the authors and reviewers discussion period. The reviewers reached a consensus of acceptance, and did not have an intense discussion during the AC and the reviewers discussion period. The AC read through the paper, review, the confidential comments to the AC, rebuttal, and the discussion. The AC supports the reviewers’ decision, and makes a decision to accept this submission. This decision was approved by the SAC.